# Toward a Learnable Artificial Intelligence Model for Aerosol Chemistry and Interactions (AIMACI) based on the Multi-Head Self-Attention Algorithm

Zihan Xia [1], Chun Zhao [1,2,3*], Zining Yang [1], Qiuyan Du [1], Jiawang Feng [1], Chen Jin [1], Jun Shi [4], Hong An [2,4]

[1]Deep Space Exploration Laboratory/School of Earth and Space Sciences/CMA-USTC Laboratory of Fengyun Remote Sensing/State Key Laboratory of Fire Science/Institute of Advanced Interdisciplinary Research on High-Performance Computing Systems and Software, University of Science and Technology of China, Hefei, China
[2]Laoshan Laboratory, Qingdao, China
[3]CAS Center for Excellence in Comparative Planetology, University of Science and Technology of China, Hefei, China.
[4]School of Computer Science and Technology, University of Science and Technology of China, Hefei, 230026, China

The manuscript for submission to *Atmospheric Chemistry and Physics.*

*Correspondence to*:  Chun Zhao (chunzhao@ustc.edu.cn)

**Abstract.** Simulating aerosol chemistry and interactions (ACI) is crucial in climate and atmospheric model, yet conventional numerical schemes are computationally intensive due to stiff differential equations and iterative methods involved. While artificial intelligence (AI) has demonstrated the potential in accelerating photochemistry simulations, it has not been applied for simulating the full ACI processes which encompass not only chemical reactions but also other processes such as nucleation and coagulation. To bridge this gap, we develop a novel Artificial Intelligence Model for Aerosol 20  Chemistry and Interactions (AIMACI), focusing initially on inorganic aerosols. Trained based on conventional scheme, it has been validated both offline and online mode (referring to whether it is coupled into three-dimensional atmospheric model). Results demonstrate that AIMACI are not only comparable to those with the conventional scheme in spatial distributions, temporal variations, and evolution of particle size distribution of main aerosol species including water content in aerosols, but also exhibits robust generalization ability, reliably simulating one month under different environmental 25  conditions across four seasons despite being trained on limited data from merely 16 days. Notably, it exhibits a $\sim 5 \times$ speedup with a single CPU and $\sim 277 \times$ speedup with a single GPU compared to conventional scheme. However, the stability of AIMACI for year-scale global simulations remain to be seen, requiring further testing. AIMACI's generalization capability and its plug-and-play nature suggest potential for future coupling into global climate models, which are expected to enhance the precision and efficiency of aerosol simulations in climate modeling that neglects or simplifies ACI processes.

# 1 Introduction

Atmospheric aerosols, which consist of a suspension of solid and liquid particles in the air, exert a profound influence on Earth's climate system and air quality (Charlson et al., 1992). Their multifaceted impacts are evident in their capacity to alter the Earth's radiation balance through the scattering and absorption of solar and longwave radiation, as well as in their role as cloud condensation nuclei that influence the formation and characteristics of clouds (Twomey, 1974; Lohmann and

Feichter, 2005; Bellouin et al., 2020; Li et al., 2022). The presence of atmospheric aerosols extends its reach to environmental well-being, with implications that span visibility, human health, and the integrity of ecological ecosystems (Pöschl, 2005; Arfin et al., 2023). Aerosol chemistry and interactions (ACI) involve a range of highly nonlinear processes, including chemical reactions and phase equilibrium, gas-particle partitioning, particle size growth, coagulation, and nucleation, which have a significant impact on the concentration of atmospheric aerosols (Zaveri et al., 2008). Numerical

models stand as indispensable analytical tools, pivotal for comprehending the aforementioned phenomena, and are instrumental in air quality management and the formulation of mitigation strategies for climate change. However, coupling ACI into these models poses a significant computational challenge (Carmichael et al., 1999; Ebel et al., 2006). As shown in Figure S1, which displays the proportion of computational time for different parts of the chemistry module, the Model for Simulating Aerosol Interactions and Chemistry (MOSAIC) scheme with just four bins for major inorganic aerosols already

accounts for 31.4% of the total computational time. This is primarily due to the requirement to solve a complex set of stiff nonlinear differential equations governing aerosol processes, coupled with the use of implicit integration schemes to ensure numerical stability (Sandu et al., 1997a, b). Furthermore, to accommodate the diverse methodologies for describing the evolution of particle size distribution (PSD), some aerosol processes may require repeated calculations (Wang et al., 2022). For example, when employing a discrete model, the coagulation collision frequency functions need to be computed for each

discrete size (Zhang et al., 2020). Consequently, the computational burden is significantly amplified. This computational intensity often creates a dilemma, as it competes with other priorities in numerical modeling, such as enhancing spatial resolution (Gu et al., 2022), recognized as helpful for minimizing uncertainties in numerical models. Numerous numerical models opt for simplified or even deactivate ACI scheme during long-term simulations, particularly in high-resolution atmospheric and climate models, introducing considerable uncertainties into the simulation results (Lee et al., 2016; Zhang et

al., 2020). Consequently, there is a pressing need to achieve rapid, accurate, and stable simulation of ACI within numerical models.

Over the past few decades, extensive research efforts have been dedicated to striking a tradeoff between the accuracy and computational efficiency in simulating ACI. Researchers have primarily approached this challenge from two distinct perspectives: one is the exploration of various methodologies for describing the evolution of PSD. For instance, in the

discrete model, the PSD is divided into discrete sizes, with calculations performed for each individual size. This approach yields the most precise results but also demands the highest computational resources (Landgrebe and Pratsinis, 1990; Zhao et

al., 2013b). The moment model, which tracks the lower-order moments of an unknown aerosol distribution, is particularly well-suited for scenarios where the size distribution is lognormal (Pratsinis, 1988). Concurrently, researchers have been engaged in employing diverse methodologies to solve the system of stiff differential equations. For example, the Multicomponent Taylor Expansion Method (MTEM) has been developed to compute activity coefficients in aqueous atmospheric aerosols (Zaveri et al., 2005). This method offers an efficient non-iterative solution for systems rich in sulfate aerosols. The Adaptive Step Time-split Euler Method (ASTEM) leverages several key characteristics of the atmospheric gas-particle partitioning, systematically reducing stiffness while preserving the integrity of the numerical solution (Zaveri et al., 2008). Despite these advancements significantly improving the computational efficiency of simulating ACI, current progress remains far from sufficient.

An alternative approach is to utilize artificial intelligence (AI) schemes to replace conventional numerical schemes in atmospheric and climate models, which could potentially bring about a transformative impact. Recent studies by Liu et al. (2021) developed an AI scheme based on a Residual Neural Network (ResNet) algorithm for simulating atmospheric photochemistry, achieving a nearly 10.6× increase in computational efficiency. However, they adopted a hybrid approach, combining the numerical scheme for radicals and oxidants with the AI scheme for volatile organic compounds (VOCs). Kelp et al. (2022) employed an online training strategy to refine an AI scheme for a simplified Super-Fast chemistry scheme (12 species) in atmospheric models, achieving stable simulations over a year with a nearly 5× speedup. Yet, this method necessitated a complex preparation process, involving training 48 separate AI model for each chemical species across four seasons. Sharma et al. (2023) developed a physics-informed AI approach to study isoprene epoxydiols in acidic aqueous aerosols over the Amazon rainforest, halving computational costs but requiring training separate AI model for each size bin. Xia et al. (2024) have taken a step further by developing an AIPC scheme leveraging the Multi-Head Self-Attention (MHSA) algorithm to simulate a full complex atmospheric photochemistry with a unified AI model. When coupled with three-dimensional (3D) numerical models, their approach not only reliably simulates the continuous spatiotemporal evolution of 74 chemical species over 15 consecutive days but also achieved a nearly 8× speedup.

While the studies discussed above have highlighted the impressive performance of AI algorithms in capturing highly nonlinear relationships between different chemical species and reproducing complex spatiotemporal distributions, to date, no AI-based scheme exists for simulating the ACI in 3D numerical models. Unlike photochemistry which only involves chemical reactions between species, the full ACI encompasses numerous other intricate processes such as nucleation, coagulation, thermodynamics. Furthermore, since the PSD of an aerosol significantly influences aerosol behavior, an accurate depiction of the evolution of PSD is as critical as the precise simulation of concentration of aerosol species. These factors collectively present a heightened challenge for the development of an AI scheme capable of simulating the full ACI. The feasibility of establishing such an AI scheme for aerosols remains an open question.

To bridge this gap, in this study, we have developed a novel Artificial Intelligence Model for Aerosol Chemistry and Interactions, termed AIMACI, which is based on the Multi-Head Self-Attention algorithm and has been online coupled with a 3D numerical atmospheric model. As the first step, this study focuses on inorganic aerosols, because the chemistry of

organic aerosols (i.e., secondary organic aerosols) still has large uncertainties and lacks a convincing numerical scheme for AI scheme to emulate, which certainly deserves further investigation in future. To validate the accuracy, stability, and computational efficiency of the AIMACI scheme, we conducted a series of experiments for both offline simulations (where AIMACI scheme was not coupled to a numerical model) and online simulations (where AIMACI scheme was coupled to a numerical model). The structure of this paper is organized as follows: Section 2 provides a detailed description of the Weather Research and Forecasting with Chemistry (WRF-Chem) model and the establishment of the AIMACI scheme. Section 3 discusses the results, and Section 4 presents the conclusion, outlining the implications of our findings for the field.

## 2 Methods

### 2.1 WRF-Chem Model and MOSAIC Scheme

In this study, we utilize the updated version of WRF-Chem developed by the University of Science and Technology of China (USTC) for conducting all simulations. This USTC version of WRF-Chem boasts additional functionalities compared to the publicly released version, including the capability to diagnose radiative forcing of aerosol species, land-surface-coupled biogenic volatile organic compound emissions, and aerosol-snow interactions (Du et al., 2020; Hu et al., 2019; Zhang et al., 2021; Zhao et al., 2013a, b, 2014, 2016).

The conventional numerical scheme of ACI adopted in this study for comparison is the MOSAIC scheme (Zaveri et al., 2008) within WRF-Chem, coupled with the CBM-Z (Carbon Bond Mechanism Version Z) photochemistry scheme (Zaveri and Peters, 1999). The MOSAIC scheme stands out for its innovative approach to address the long-standing issues in solving the dynamic partitioning of semivolatile inorganic gases ($HNO_3$, $HCl$, and $NH_3$) to size-distributed atmospheric aerosol particles. It has been validated against a benchmark model version utilizing a rigorous solver for the integration of stiff differential equations, demonstrating both computational efficiency and high fidelity (Zaveri et al., 2008). The MOSAIC scheme used in this study treats all the major aerosol species important at urban, regional, and global scales, including sulfate ($SO_4^{2-}$), nitrate ($NO_3^-$), chloride ($Cl^-$), carbonate ($CO_3^{2-}$), ammonium ($NH4^+$), sodium ($Na^+$), calcium ($Ca^{2+}$), black carbon (BC), organic carbon (OC), other inorganic mass (OIN), mineral dust, methanesulfonic acid (MSA) and liquid water content of aerosol (Water). The chemical reactions among various species are detailed in Zaveri et al. (2008). Currently, only a subset of sulfate and nitrate chemical reactions are considered, such as the interactions between sulfuric acid, nitric acid, and ions like calcium, sodium, and chloride, which lead to the formation of sulfates and nitrates. For more complex chemical reactions, such as the different heterogeneous formation pathways of sulfates in aerosol liquid water (e.g., oxidation of dissolved S(IV) by $H_2O_2$, $O_3$, $NO_2$ and $O_2$ catalyzed by transition metal ions (TMI) in aerosol water, (Ruan et al., 2022)), will be incorporated in the future. The MOSAIC scheme employs a sectional approach, dividing the PSD into 4 discrete size bins in this study. The first, second, third, and fourth size bins are set to be 0.039~0.156, 0.156~0.625, 0.625~2.5, and 2.5~10.0

µm in diameter, respectively. In addition, it further considers the impact of marine biogenic sources of dimethyl sulfide on atmospheric aerosols and aqueous aerosol processes.

## 2.2 Learnable AIMACI Scheme

### 2.2.1 Scheme Construction

Previous attempts into the substitution of conventional numerical schemes with AI schemes have predominantly utilized simple AI algorithms, such as Random Forest Regression (Keller and Evans, 2019). This preference stems from the complex challenge of coupling sophisticated AI algorithms, often written in Python, with numerical models coded in Fortran. While some studies have explored the use of advanced AI algorithms, such as Fully Connected Neural Networks (Sharma et al., 2023) and Residual Neural Networks (Kelp et al., 2018, 2020, 2022; Liu, 2021; Wang et al., 2022), these have occasionally

encountered difficulties when dealing with high-dimensional input variables and have demonstrated limitations in accurately simulating highly nonlinear systems (Xia et al., 2024).

In this study, for the first time, we attempt to use an AI algorithm to emulate the sophisticated scheme of ACI (i.e., the MOSAIC scheme), which involves a range of highly nonlinear processes, including chemical reactions and phase equilibrium, gas-particle partitioning, particle size growth, coagulation and nucleation (Zaveri et al., 2008). Given the

complexity of these interactions, there is a clear need for AI algorithms with superior representational capacity for nonlinear systems. Xia et al. (2024) have highlighted that the MHSA algorithm excels in capturing the intricate chemical relationships among different species. It offers not only high simulation accuracy and computational efficiency but is also less susceptible to the increase in the number of chemical species.

Therefore, in this study, we introduce an innovative application of MHSA algorithm in the field of ACI simulation

through the development of the Artificial Intelligence Model for Aerosol Chemistry and Interactions (AIMACI). Although the MHSA algorithm has been instrumental in the advancement of state-of-the-art transformer models in domains such as Natural Language Processing, Computer Vision and Weather Forecast, as evidenced by the seminal works of Vaswani et al. (2017), Liu et al. (2021), and Bi et al. (2023), its utilization in ACI simulation represents a new horizon. The algorithm's ability to globally attend to input variables and conduct parallel computations across multiple heads is pivotal in tackling the

challenges posed by the curse of dimensionality, capturing complex interdependencies, and significantly enhancing computational efficiency.

Figure 1 illustrates the integration of AIMACI scheme in our hybrid atmospheric model with physics and AI schemes (physics-AI hybrid model), and also provides a schematic representation of the AI model architecture that is utilized within the AIMACI scheme. The AI model architecture is intricately designed with three principal components, each serving a

distinct function in the simulation process: (1) Input Embedding Layer: This initial layer receives meteorological variables and chemical species as input features. The input embedding layer is designed as a fully connected layer, which maps the raw input data into a higher-dimensional space where interdependencies between variables can be more effectively captured.

(2) Integrator: As the core of the AI model, it is composed of 2 identical blocks, each of which contains two sub-layers: a multi-head attention layer and a feed-forward layer. We apply residual connections around each of these two sub-layers, followed by layer normalization. This integrator is responsible for learning the complex and high nonlinear processes of ACI within the data and integrate them over time. (3) Output Representation Layer: Following the integrator, it also implemented as a fully connected layer. This layer translates the processed information from the integrator into chemical concentrations, providing the output targets for the simulation. Furthermore, the AI model is complemented by pre-processing and post-processing steps, such as min-max normalization, to constitute the comprehensive AIMACI scheme.


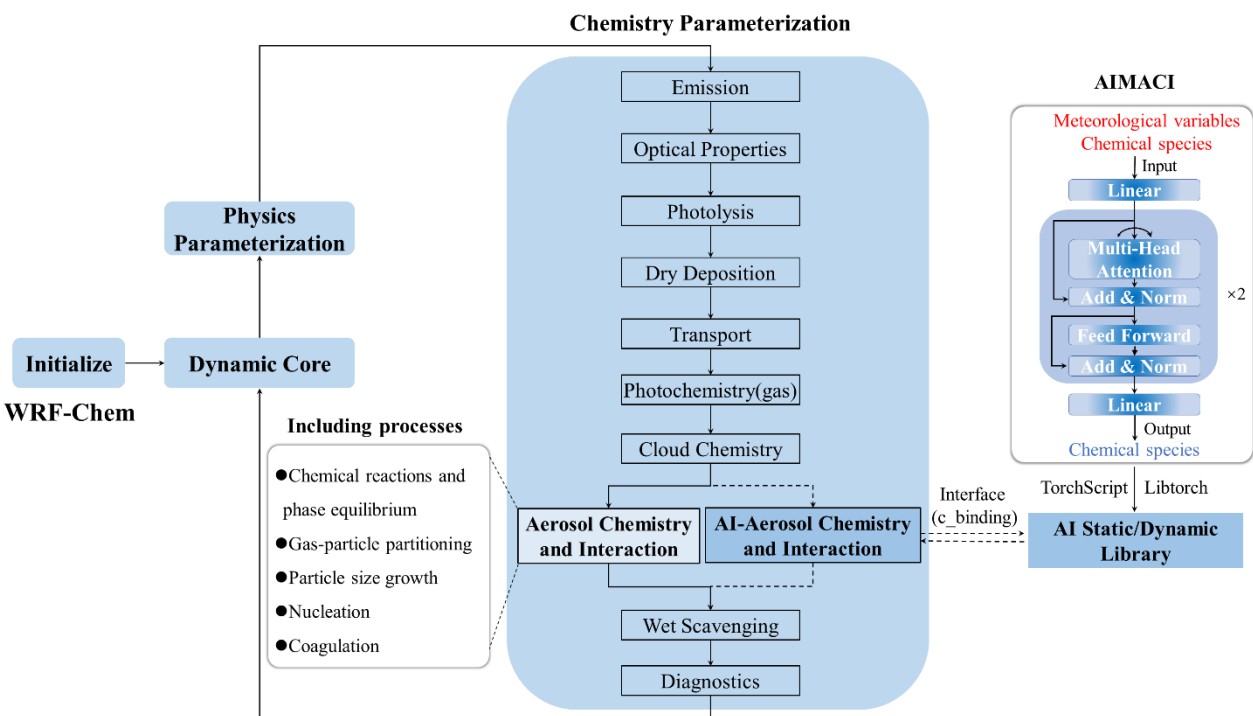


**Figure 1:** The Artificial Intelligence Model for Aerosol Chemistry and Interactions (AIMACI) in the Weather Research and Forecasting with Chemistry (WRF-Chem). The trained AIMACI is packaged into a static/dynamic library using TorchScript and Libtorch, and can be called by WRF-Chem through an interface to replace the aerosol chemistry and interactions numerical scheme, while the remaining processes maintain the original numerical scheme.

## 2.2.2 Training and Testing Procedure

To generate the training, validation, and test datasets, we conducted the WRF-Chem simulations over East China, spanning the period from 2019-03-01 00:00 UTC to 2019-03-19 23:00 UTC. The simulation result was segmented as follows: the initial 16 days from 2019-03-02 00:00 UTC, were designated as the training set, the penultimate day served as the validation set, and the final day constituted the test set. The simulation was configured with a 0.2° horizontal resolution, covering $140 \times 105$ grid cells within the geographical bounds of 107.1° E to 127.9° E and 19.7° N to 47.5° N, and featured 49 vertical layers extending up to 50 hPa. A dynamic time step of 2 minutes and a chemical time step of 1 hour were employed. For emission and meteorological field, we used the Multi-resolution Emission Inventory for China (MEIC) at 0.25° x 0.25° resolution for 2019 (Li et al., 2017a; Li et al., 2017b), and the NCEP final reanalysis (FNL) data with a 1° x 1° resolution and 6-hour temporal resolution within the simulation domain. Concentrations of aerosol and gas species pertinent to gas-particle partitioning were recorded hourly, along with key meteorological variables influencing chemistry: temperature, pressure, air density, and water vapor mixing ratio. A comprehensive list of variables used for training the AIMACI scheme is presented in Table 1. Due to computational cost considerations, the period of the training dataset is not very long; however, the volume of training samples is large due to the hourly chemical time step and the fine spatial resolution of our simulation. With 140 by 105 grid cells, 49 vertical layers, and 24 hours in a day, the total number of training samples amounts to 276,595,200 (140x105x49x24x16), reaching the hundred million scale. This large dataset provides a rich and diverse set of samples for training, ensuring that the AI model does not suffer from a lack of convergence due to insufficient data.

In the training of the AI model we built from scratch, each training sample included 65 input features (4 meteorological variables, 5 gas species, and 14 aerosol species with 4 size bins) and 61 output targets (5 gas species and 14 aerosol species with 4 size bins). All features and targets underwent min-max normalization to standardize the data. We employed the PyTorch deep learning framework for model training, with a batch size of 2048, an initial learning rate of 0.001, and the Adam optimization algorithm. The Mean Squared Error (MSE) was used as the loss function. To optimize the training process, we implemented a learning rate decay strategy using the ReduceLROnPlateau scheduler, along with an early stopping mechanism after 10 consecutive epochs without improvement in the validation loss. All other hyperparameters not mentioned are kept at their default values. For this study, we trained the model using three GPUs for approximately three days, and the model achieved optimal performance at epoch 32.

After training, the AIMACI scheme was packaged into a static library and then flexibly coupled into WRF-Chem, utilizing TorchScript and Libtorch tools officially provided by PyTorch. Compared to using the third-party libraries like CFFI (C Foreign Function Interface for Python), our coupling approach offers several advantages. Firstly, from a performance perspective, TorchScript and Libtorch tools are part of the PyTorch ecosystem, optimized for running AI models in C++ environments, thus providing faster execution than third-party libraries. Secondly, in terms of compatibility, LibTorch supports cross-platform deployment, offering more flexibility than third-party libraries. Lastly, regarding ease of

use, the official PyTorch documentation provides comprehensive examples on how to package PyTorch-trained AI models into static or dynamic libraries using TorchScript and Libtorch. This significantly simplifies the process, as we mainly need
to write an interface to call the AI model, contrasting with third-party libraries, which would necessitate writing the entire codebase from scratch. Therefore, this coupling approach minimizes alterations to the original codebase and offers a lightweight, adaptable, and easily plug-and-play solution. It is capable of encapsulating a wide range of complex AI algorithms and coupling them with diverse atmospheric and climate models.

Furthermore, to comprehensively evaluate the performance of the AIMACI scheme, we conducted a series of additional
experiments in both offline and online mode, which are detailed in Table 2. These experiments included: (1) Offline simulations were performed on the test dataset without coupling AIMACI scheme into WRF-Chem, treating AIMACI as a standalone box model to assess its single-step performance; (2) Online simulations were conducted for a 10-day continuous period outside the training phase, following the coupling of the AIMACI scheme into WRF-Chem, where it is invoked at each chemical time step and interacts with other model processes to create a dynamic and iterative aerosol simulation; (3) A
month-long online continuous simulation was carried out under different environmental conditions across all four seasons to evaluate the AIMACI scheme's generalization ability. To assess the potential impact of the AIMACI scheme on climate research applications, we evaluated statistical indicators for all species and a focused examination of spatial distributions, temporal series, and the evolution of particle size distribution (PSD) for four representative aerosol species. The four selected species include sulfate, nitrate, liquid water content in aerosols, and aerosol number concentration. Sulfate, mainly
from fossil fuel emissions, contributes to acid rain, aerosol formation, and aerosol-cloud interactions (Calvert et al., 1985; Fuzzi et al., 2015; Penkett et al., 1979). Nitrate, formed from nitrogen oxides, is a key aerosol component that affects air quality and ecosystems. (Parrish et al., 2012; Saiz-Lopez et al., 2017). Liquid water content in aerosols is important for understanding how particles contribute to cloud formation and precipitation. (Hodas et al., 2014; Liu et al., 2019; Nguyen et al., 2016; Wu et al., 2018). Aerosol number concentration serves as a key metric for assessing aerosol loading and its direct
impact on visibility, radiation balance, and climate feedback mechanisms (Spracklen et al., 2010). Collectively, these four species offer a holistic perspective on the multifaceted role of aerosols in atmospheric processes.

**Table 1:** Input and output variables of Artificial Intelligence Model for Aerosol Chemistry and Interactions (AIMACI).

| Type | Input variables | Output variables |
|---|---|---|
| Meteorological variables | temperature | - |
| | air density | - |
| | pressure | - |
| | water vapor mixing ratio | - |
| Gas species | $H_2SO_4$ | $H_2SO_4$ |
| | $HNO_3$ | $HNO_3$ |

| | | NH3 | NH$_3$ |
| --- | --- | --- | --- |
| | | HCL | HCL |
| | | MSA | MSA |
| Aerosol species | | SO$_4^{2-}$ [Size:1-4] | SO$_4^{2-}$ [Size:1-4] |
| | | NO$_3^-$ [Size:1-4] | NO$_3^-$ [Size:1-4] |
| | | NH$_4^+$ [Size:1-4] | NH$_4^+$ [Size:1-4] |
| | | Na$^+$ [Size:1-4] | Na$^+$ [Size:1-4] |
| | | Cl$^-$ [Size:1-4] | Cl$^-$ [Size:1-4] |
| | | MSA[Size:1-4] | MSA[Size:1-4] |
| | | Water [Size:1-4] | Water [Size:1-4] |
| | | Num [Size:1-4] | Num [Size:1-4] |
| | | OIN [Size:1-4] | OIN [Size:1-4] |
| | | DUST [Size:1-4] | DUST [Size:1-4] |
| | | OC [Size:1-4] | OC [Size:1-4] |
| | | BC [Size:1-4] | BC [Size:1-4] |
| | | Ca$^{2+}$ [Size:1-4] | Ca$^{2+}$ [Size:1-4] |
| | | CO$_3^{2-}$ [Size:1-4] | CO$_3^{2-}$ [Size:1-4] |

**Table 2:** Numerical experiments conducted in this study.

| Number | ACI Scheme | Period (Hourly) | Type |
| --- | --- | --- | --- |
| EXP 0 | MOSAIC | Mar 1$^{st}$ ~ Mar 19$^{th}$ (Train: Mar 2$^{nd}$ ~ Mar 17$^{th}$, Validation: Mar 18$^{th}$, Test: Mar 19$^{th}$) | Online Continuous Simulation |
| EXP 1 | AIMACI | Mar 19$^{th}$ | Offline Single-step Simulation |
| EXP 2&3 | MOSAIC & AIMACI | Mar 20$^{th}$ ~ Mar 30$^{th}$ | Online Continuous Simulation |
| EXP 4&5 | MOSAIC & AIMACI | Jan, Apr, Jul, Oct (1 month) | Online Continuous Simulation |

## 2.3 Evaluation metric

In this research, a comprehensive evaluation of the AIMACI scheme's effectiveness was conducted utilizing three recognized statistical measures. For every species examined, the calculation of the Pearson correlation coefficient ($R^2$), the Root Mean Square Error (RMSE), the Normalized Mean Bias (NMB), the absolute error (AE) and relative error (RE) was performed.

$$\uparrow (R^2) = \frac{\left( \sum_{i=1}^{N_{lat}} \sum_{j=1}^{N_{lon}} L(i)\left(c_{i,j} - \bar{c}\right)\left(\hat{c}_{i,j} - \bar{\hat{c}}\right)\right)^2}{\sum_{i=1}^{N_{lat}} \sum_{j=1}^{N_{lon}} L(i)\left(c_{i,j} - \bar{c}\right)^2 \times \sum_{i=1}^{N_{lat}} \sum_{j=1}^{N_{lon}} L(i)\left(\hat{c}_{i,j} - \bar{\hat{c}}\right)^2} \tag{1}$$

$$\downarrow RMSE = \sqrt{\frac{\sum_{i=1}^{N_{lat}} \sum_{j=1}^{N_{lon}} L(i)\left(\hat{c}_{i,j} - c_{i,j}\right)^2}{N_{lat} \times N_{lon}}} \tag{2}$$

$$\downarrow NMB = \frac{\sum_{i=1}^{N_{lat}} \sum_{j=1}^{N_{lon}} L(i)(\hat{c}_{i,j} - c_{i,j})}{\sum_{i=1}^{N_{lat}} \sum_{j=1}^{N_{lon}} L(i)c_{i,j}} \tag{3}$$

$$\downarrow AE = \hat{C} - C \tag{4}$$

$$\downarrow RE = \frac{\hat{C} - C}{C} \tag{5}$$

In above, $L(i) = N_{lat} \times \frac{\cos \varphi_i}{\sum_{i=1}^{N_{lat}} \cos \varphi_i}$ is the weight at latitude $\varphi_i$, $\hat{C}$ denotes the concentration simulated by the AIMACI scheme, $C$ denotes the concentration simulated by the MOSAIC scheme, $\uparrow$ denotes higher values are better, $\downarrow$ denotes lower values are better.

## 2.4 Computational configuration

A primary incentive for coupling AI schemes into atmospheric and climate models is the pursuit of substantial computational acceleration. However, such acceleration is not inherently guaranteed, as demonstrated by Keller et al. (2019). Consequently, it is imperative to meticulously compare the temporal expenditure of AI schemes against those of traditional numerical schemes.

In this study, we undertook a comparative analysis of the computational time required by the numerical scheme and the AIMACI scheme for simulating ACI in 720,300 discrete grid cells, which roughly corresponds to a global simulation at 2.5° × 2.5° horizontal resolution with 72 vertical layers. To ensure a holistic and unbiased assessment of the speedup achieved, we measured the computational time by averaging the duration of 24 consecutive daily simulations. Both schemes were tested utilizing a single CPU core and additionally evaluated the AIMACI scheme with a GPU-accelerated scenario using a single GPU. The computational hardware employed in our tests consisted of an Intel Xeon Scalable 8358 CPU and an NVIDIA A100-80G GPU.

## 3 Results

 **3.1 Offline Single-step Simulations with the AIMACI Scheme**

Before coupling the AIMACI scheme with the 3D numerical model WRF-Chem for continuous simulation, we first evaluated its performance on a test dataset that was separate from the training data. The test dataset, as detailed in previous sections, comprises a series of 3D spatial outcomes taken at 24-hourly intervals on March 19, 2019. It provides representative samples that span a wide range of meteorological conditions and species concentrations. The evaluation on this dataset provides insight into the AIMACI scheme's performance in various atmospheric conditions. Table 2 presents the statistical metrics for all simulated species, offering a comprehensive assessment of the scheme's simulation capabilities.

The results are promising, with an average R² of 0.98 for all 61 evaluated species. This high degree of correlation indicates a strong consistency between the simulations using the AIMACI scheme and the MOSAIC scheme (hereinafter referred to as the numerical scheme). The average NMB for these species is 3.02%, reflecting only a slight deviation from the numerical scheme's outcomes and highlighting the AIMACI scheme's impressive accuracy in simulating ACI. However, as shown in Table 3, some species still exhibit relatively poorer statistical indicators compared to others, such as carbonates. To delve deeper into this observation, we have plotted the frequency histograms of the concentration distributions for all species in the test data (Figure S2). Our analysis revealed that species with skewed concentration distributions, particularly those where more than 99% of the values are close to zero, tend to exhibit poorer statistical indicators. However, this does not signify that the AIMACI scheme has entirely forfeited its predictive capability. As demonstrated in Figure S3, which illustrates the simulated carbonate concentrations in the 0.625–2.5 µm particle size range ($CO_3\_a03$), the AIMACI scheme continues to perform well in predicting concentration changes in high-value regions. The poorer statistical indicators are primarily attributed to the challenge of accurately forecasting the very low values that are close to zero.

**Table 3:** Statistical metrics on the test dataset of Artificial Intelligence Model for Aerosol Chemistry and Interactions (AIMACI) (The RMSE of different species has different unit: aerosol (µg/kg), num (kg-1), gas (ppmv)).

| Number | Variable | $R^2$ | RMSE | NMB (%) | Number | Variable | $R^2$ | RMSE | NMB (%) |
|---|---|---|---|---|---|---|---|---|---|
| 1 | $H_2SO_4$ | 0.97 | 2.99E-07 | -2.10 | 32 | Ca_a02 | 0.95 | 2.32E-05 | -2.87 |
| 2 | $HNO_3$ | 1.00 | 3.61E-05 | 0.19 | 33 | $CO_3\_a02$ | 0.61 | 2.90E-05 | 28.40 |
| 3 | $NH_3$ | 1.00 | 4.84E-05 | 3.49 | 34 | $SO_4\_a03$ | 0.94 | 2.90E-02 | 0.80 |
| 4 | HCL | 1.00 | 9.68E-06 | -0.41 | 35 | $NO_3\_a03$ | 1.00 | 2.68E-02 | 0.89 |
| 5 | MSA | 0.82 | 1.03E-09 | 0.12 | 36 | $NH_4\_a03$ | 1.00 | 5.37E-03 | 0.48 |
| 6 | $SO_4\_a01$ | 1.00 | 1.14E-02 | 0.16 | 37 | Na_a03 | 1.00 | 2.01E-03 | 2.44 |
| 7 | $NO_3\_a01$ | 1.00 | 4.90E-02 | -0.43 | 38 | Cl_a03 | 1.00 | 4.98E-03 | 1.29 |

| | | | | | | | | | |
|---|---|---|---|---|---|---|---|---|---|
| 8 | NH$_4$_a01 | 1.00 | 1.44E-02 | -0.45 | 39 | MSA_a03 | 1.00 | 2.94E-06 | 2.85 |
| 9 | Na_a01 | 1.00 | 6.10E-06 | 0.32 | 40 | Water_a03 | 0.99 | 4.11E-01 | 2.03 |
| 10 | Cl_a01 | 0.99 | 2.52E-03 | -1.42 | 41 | Num_a03 | 1.00 | 1.05E+05 | 0.45 |
| 11 | MSA_a01 | 1.00 | 1.78E-05 | 5.70 | 42 | OIN_a03 | 1.00 | 1.86E-02 | 6.34 |
| 12 | Water_a01 | 1.00 | 5.60E-01 | 0.46 | 43 | DUST_a03 | 1.00 | 1.30E-01 | 2.30 |
| 13 | Num_a01 | 1.00 | 4.45E+07 | -0.10 | 44 | OC_a03 | 1.00 | 1.37E-02 | -0.45 |
| 14 | OIN_a01 | 1.00 | 7.98E-03 | 4.36 | 45 | BC_a03 | 1.00 | 2.61E-03 | -0.53 |
| 15 | DUST_a01 | 0.95 | 2.47E-04 | 3.80 | 46 | Ca_a03 | 1.00 | 5.28E-04 | 2.62 |
| 16 | OC_a01 | 1.00 | 4.97E-03 | 0.80 | 47 | CO$_3$_a03 | 0.90 | 1.42E-03 | 88.17 |
| 17 | BC_a01 | 1.00 | 1.49E-03 | -0.64 | 48 | SO$_4$_a04 | 1.00 | 7.25E-04 | 0.27 |
| 18 | Ca_a01 | 0.95 | 9.93E-07 | -0.49 | 49 | NO$_3$_a04 | 0.99 | 5.11E-02 | 0.75 |
| 19 | CO$_3$_a01 | 0.79 | 5.02E-12 | 0.09 | 50 | NH$_4$_a04 | 0.98 | 5.62E-03 | -3.21 |
| 20 | SO$_4$_a02 | 1.00 | 4.16E-02 | 0.38 | 51 | Na_a04 | 1.00 | 9.19E-03 | 4.13 |
| 21 | NO$_3$_a02 | 1.00 | 7.04E-02 | 0.69 | 52 | Cl_a04 | 1.00 | 1.13E-02 | 3.28 |
| 22 | NH$_4$_a02 | 1.00 | 2.31E-02 | 0.51 | 53 | MSA_a04 | 0.79 | 2.00E-05 | 7.93 |
| 23 | Na_a02 | 1.00 | 1.65E-04 | 0.10 | 54 | Water_a04 | 0.99 | 1.35E+00 | -3.32 |
| 24 | Cl_a02 | 1.00 | 4.63E-03 | -1.25 | 55 | Num_a04 | 1.00 | 5.89E+03 | 2.86 |
| 25 | MSA_a02 | 1.00 | 1.51E-05 | 3.05 | 56 | OIN_a04 | 1.00 | 2.27E-02 | 3.93 |
| 26 | Water_a02 | 0.99 | 1.32E+00 | 1.58 | 57 | DUST_a04 | 1.00 | 8.85E-01 | -4.84 |
| 27 | Num_a02 | 1.00 | 9.20E+06 | 0.61 | 58 | OC_a04 | 0.96 | 4.72E-04 | -1.38 |
| 28 | OIN_a02 | 1.00 | 4.49E-02 | -1.80 | 59 | BC_a04 | 0.95 | 1.09E-04 | 5.14 |
| 29 | DUST_a02 | 1.00 | 5.25E-03 | -0.39 | 60 | Ca_a04 | 1.00 | 3.24E-03 | 1.51 |
| 30 | OC_a02 | 1.00 | 4.23E-02 | 2.67 | 61 | CO$_3$_a04 | 0.95 | 1.39E-02 | 15.00 |
| 31 | BC_a02 | 1.00 | 1.12E-02 | -2.54 | Average R$^2$: 0.98 | | | Average NMB: 3.02% | | |

Figure 2 presents the data density and distribution for column concentration of the four key aerosol species selected in Section 2.2.2. The results from the numerical scheme simulations indicate that sulfate, nitrate, and liquid water content of aerosol exhibit higher column concentrations within 0.156 to 0.625 μm (size bin 2), whereas the number concentration is notably larger within 0.039 to 0.156 μm (size bin 1). This indicates that, despite the greater number of smaller particles in size bin 1, their overall contribution to the total mass is less significant due to their lower individual mass compared to the larger particles in size bin 2. The AIMACI scheme effectively captures these nuanced aerosol characteristics, as corroborated by the exemplary R² values of 1.0 depicted in Figure 2, which underscore the scheme's fidelity in modeling aerosol behavior across various particle sizes.

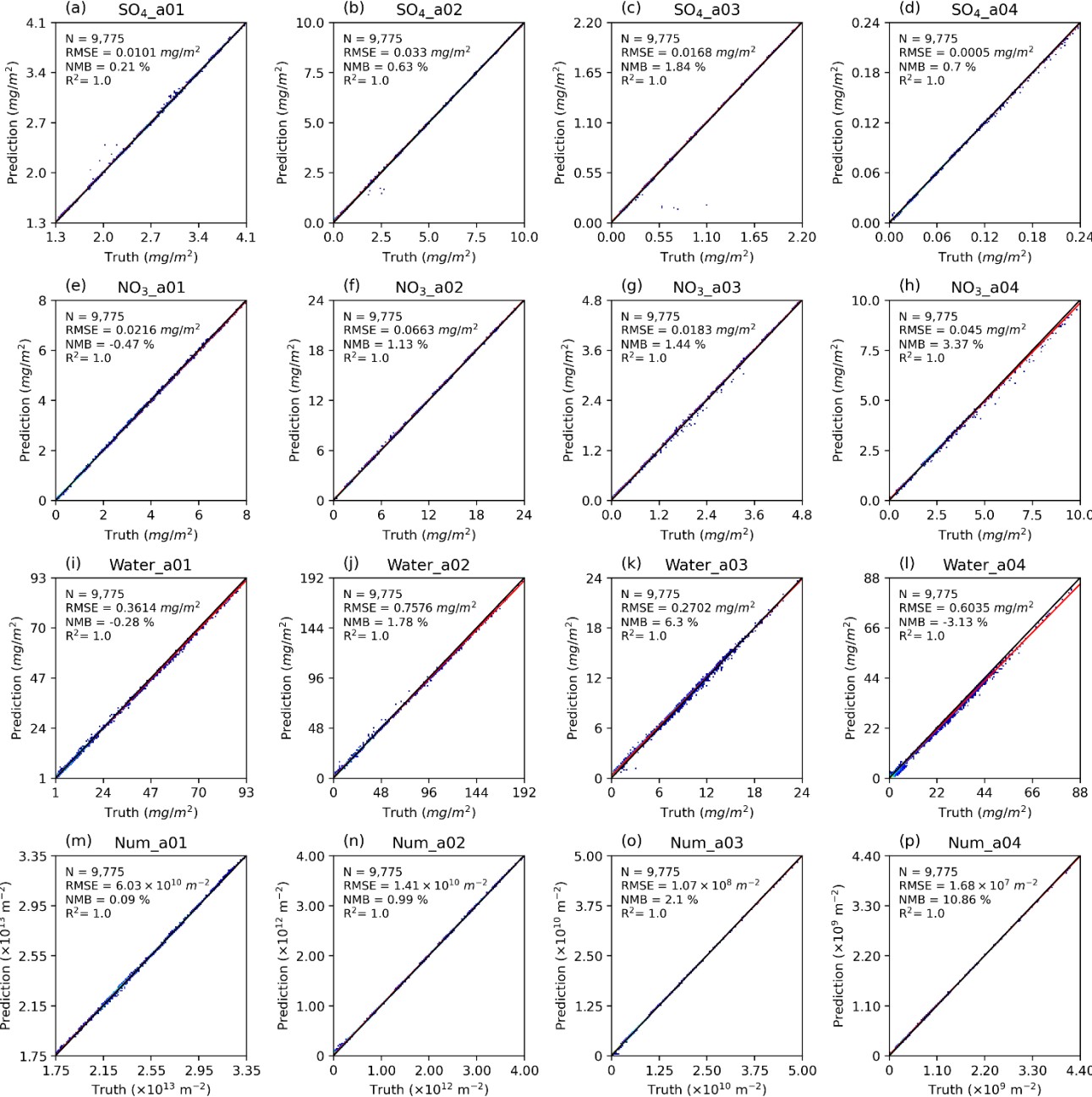

**Figure 2**: The density plot of column concentration of four key aerosol species (sulfate (SO$_4^{2-}$), nitrate (NO$_3^-$), liquid water content of aerosol (Water), and number concentration of aerosol (Num)) simulated by the AIMACI scheme on the test dataset. The results are calculated by covering the region spanning from 109.1°E to 125.9°E and from 22.1°N to 44.9°N,

with the data being averaged across the time dimension. The black line is the identical line (y=x), and the red line is fitted line.

## 3.2 Online Multi-step Simulations with the AIMACI Scheme

Unlike offline single-step simulations, coupling the AIMACI scheme into 3D numerical models to form a physics-AI hybrid model for continuous simulation entails interactions and feedback with numerous other processes. Consequently, a thorough evaluation of the AIMACI scheme's online simulation performance is essential. We focus on its performance across three critical dimensions: (1) Stable and Accurate Simulation Capability: The AIMACI scheme should accurately reproduce the spatiotemporal and size distribution of various aerosol species without rapid accumulation of errors during the

simulation process. (2) Robust Generalization Ability: The AIMACI scheme should be applicable to scenarios beyond the training data, such as different seasons, demonstrating its robustness in a variety of environmental conditions. (3) High Computational Efficiency: Compared to the conventional numerical scheme, the AIMACI scheme should offer enhanced computational efficiency, which is vital for high-resolution, long-term simulations. Although these requirements are often challenging to satisfy simultaneously, achieving these benchmarks is crucial for leveraging the full potential of the AIMACI

scheme in advancing our understanding of aerosol interactions and their impact on climate change.

### 3.2.1 Stable and Accurate Simulation Capability

     Coupling AI schemes into numerical models for stable and accurate simulations across multiple time steps has long been a formidable challenge. While the simulation errors for individual species at each time step may be minimal, they can accumulate over multiple time steps, and may even spread to other species and physical-chemical processes, leading to

chaotic simulation outcomes at the end. Typically, simulations with sophisticated aerosol processes at high-resolution, such as those in WRF-Chem, are limited to a few weeks due to computational costs. In this study, we conducted a 10-day continuous simulation from 2019-03-20 00:00 UTC to 2019-03-30 00:00 UTC to evaluate the performance of the AIMACI scheme in a coupled mode.

     Figure 3 illustrates the spatial distribution of sulfate column concentrations across different size bins at the end of the

10-day continuous simulation (i.e., 2019-03-30 00:00 UTC). The figure also tracks the temporal evolution of the RMSE throughout the simulation period. The results reveal that the high-value areas of sulfate column concentrations for different particle sizes exhibit a hook-like structure, stretching from the Yangtze River Economic Belt northeastward to the northeastern regions of China. The distinct patterns may be attributed to the complex interplay of meteorological conditions, emission sources, and atmospheric transport processes. The sulfate column concentrations are predominantly concentrated

within the 0.156 to 0.625 μm (size bin 2), with relatively lower column concentrations in the 2.5 to 10 μm (size bin 4), which is consistent with the findings in Figure 2. From the absolute error figures, it is observed that for each particle size, AIMACI tends to underestimate the higher concentration regions and overestimate lower values, particularly those near zero. This

phenomenon may be related to the low proportion of relatively high values in the training dataset and the use of RMSE as the loss function in our model training, which may bias the model towards predicting the mean. Additionally, the results shown are from the last time step of a 10-day continuous simulation, and the simulation errors could be influenced not only by the biases of a single simulation instance but also by potential inaccuracies in the inputs at that time step. Further exploration is necessary to reach a precise conclusion.

In the development of our AIMACI scheme, we faced a bifurcation of choices: In the development of our AIMACI scheme, we faced a bifurcation of choices: whether to input all features of a single grid point and predict for that grid point individually, followed by iterating through all grid points, or to input all features for the entire 3D grid space simultaneously and predict for the entire 3D space at once. Most current AI large models such as Pangu (Bi et al., 2023) and Fengwu (Chen et al., 2023), opt for the latter approach, which inevitably requires the use of convolutional networks. However, in designing the AIMACI scheme, we chose the former method, aligning with the approach taken by numerical models. This method has the advantage of significantly increasing the training sample volume, as each grid point at a given moment constitutes a sample, and it avoids the use of convolutional neural networks, leading to a substantial reduction in computational costs. Moreover, this grid-based AI scheme is versatile, capable of being applied to simulations of regions of any size, without constraints imposed by the size of the training area. However, this approach also presents a challenge in accurately simulating spatial distributions, given the potential for error propagation from neighboring grid points due to physical processes like transport. In Figure 3, the AIMACI scheme has effectively captured the overall spatial distribution of sulfate column concentrations across various particle sizes, with $R^2$ values all exceeding 0.88, even after a 10-day simulation. Although there are instances of underestimation or overestimation, the RMSE time series indicates that the RMSE values remain small throughout the entire simulation period, highlighting the scheme's stability and accuracy. Furthermore, we calculated the slope of RMSE time series across different simulation stages for four aerosol size bins. It reveals that the RMSE trends are not uniform across different simulation stages, and even within the same stage, the RMSE trends for different species vary. This indicates that the simulation error for each species is not consistently increasing; there are instances where it decreases. Furthermore, not all species exhibit a simultaneous increase in simulation error; some species show an increase while others show a decrease. This complex error variation may be related to the online simulation approach, as the aerosol concentrations simulated by the AIMACI scheme are subject to other processes in the numerical model such as dry deposition, wet scavenging.

Figure 4 presents a comparison of the zonal average total concentrations (summed across all size bins) of sulfate and nitrate, simulated by both the numerical scheme and the AIMACI scheme, with results averaged over the entire 10-day simulation period. Observations from Figure 4a and 4c indicate that high concentration zones for both sulfate and nitrate are predominantly situated between 25°N and 40°N, coinciding with the latitude range of the Yangtze River Economic Belt. This distribution pattern is likely influenced by the significant anthropogenic emissions in this area. Through turbulent and convective transport processes, sulfate and nitrate from lower altitudes are transported to higher altitudes, with concentrations gradually diminishing with increasing altitude. In Figures 4b and 4d, the AIMACI scheme exhibits a notable

alignment with the outcomes from the numerical scheme, as evidenced by the R² values, which are exceptionally high at 0.99 for both sulfate and nitrate. The Root Mean Square Error (RMSE) values are 0.10 µg/kg for sulfate and 0.48 µg/kg for nitrate, suggesting that the discrepancies are minimal, further supporting the AIMACI scheme's accuracy. To provide additional

insights, we have also included plots for both the absolute and relative errors. The absolute error distribution shows larger errors near the surface and smaller ones at higher altitudes, while the relative error distribution follows the opposite trend, with lower relative errors near the surface and higher relative errors at greater altitudes. This pattern is particularly prominent for nitrate, which, compared to sulfate, has significantly lower concentrations at higher altitudes—often by one or two orders of magnitude, approaching zero.

Figure 5 illustrates the ability of the AIMACI scheme to reproduce temporal variations of surface total concentrations of four key aerosol species. These results represent the calculated averages for the Yangtze River Delta region, a crucial urban agglomeration in China, spanning the coordinates 119.1°E to 121.9°E and 30.1°N to 31.9°N. Throughout the simulation period, sulfate concentrations primarily fluctuate within the range of 0 to 6 µg/kg, while nitrate concentrations exhibit a broader variability, predominantly ranging from 0 to 20 µg/kg. Notably, all four key aerosol species experience

several instances of abrupt concentration spikes and declines. For instance, between the 11th and 30th hour of the simulation, the liquid water content of aerosol experiences a dramatic increase from 32.86 µg/kg to 263.47 µg/kg, followed by a sharp decrease to 28.72 µg/kg. Despite these pronounced fluctuations, the AIMACI scheme adeptly reproduces these features without introducing systematic bias, achieving R² values larger than 0.97.

As discussed above, the AIMACI scheme 's proficiency in simulating the spatiotemporal distribution and variation trend

of different aerosol species is well-established. However, accurately reproducing the evolution of the aerosol PSD is equally vital, given the significant role particle size plays in dictating the interactions of aerosols with clouds and radiation`, which are pivotal for atmospheric processes. Figure 6 presents the PSD and frequency distribution for the surface concentrations of four key aerosol species simulated by the AIMACI scheme. The frequency distributions of sulfate and nitrate surface concentrations exhibit a relatively uniform pattern, whereas the liquid water content and number concentration of aerosols

display extreme values, leading to pronounced skewness in their distributions. The AIMACI scheme accurately captures these distributions, although it tends to overestimate minimal concentration values that approach zero across different particle sizes, a minor deviation that could be addressed in future refinements. Notably, there are significant differences in the PSD among the aerosol species. Sulfate and nitrate concentrations peak within the 0.156 to 0.625 µm (size bin 2), whereas the liquid water content of aerosols is most concentrated in the 2.5 to 10.0 µm (size bin 4) and the number

concentration of aerosols is predominantly found in the 0.039 to 0.156 µm (size bin 1). These findings highlight the complexity of accurately modeling PSD and the AIMACI scheme's commendable performance in reproducing these intricate patterns.

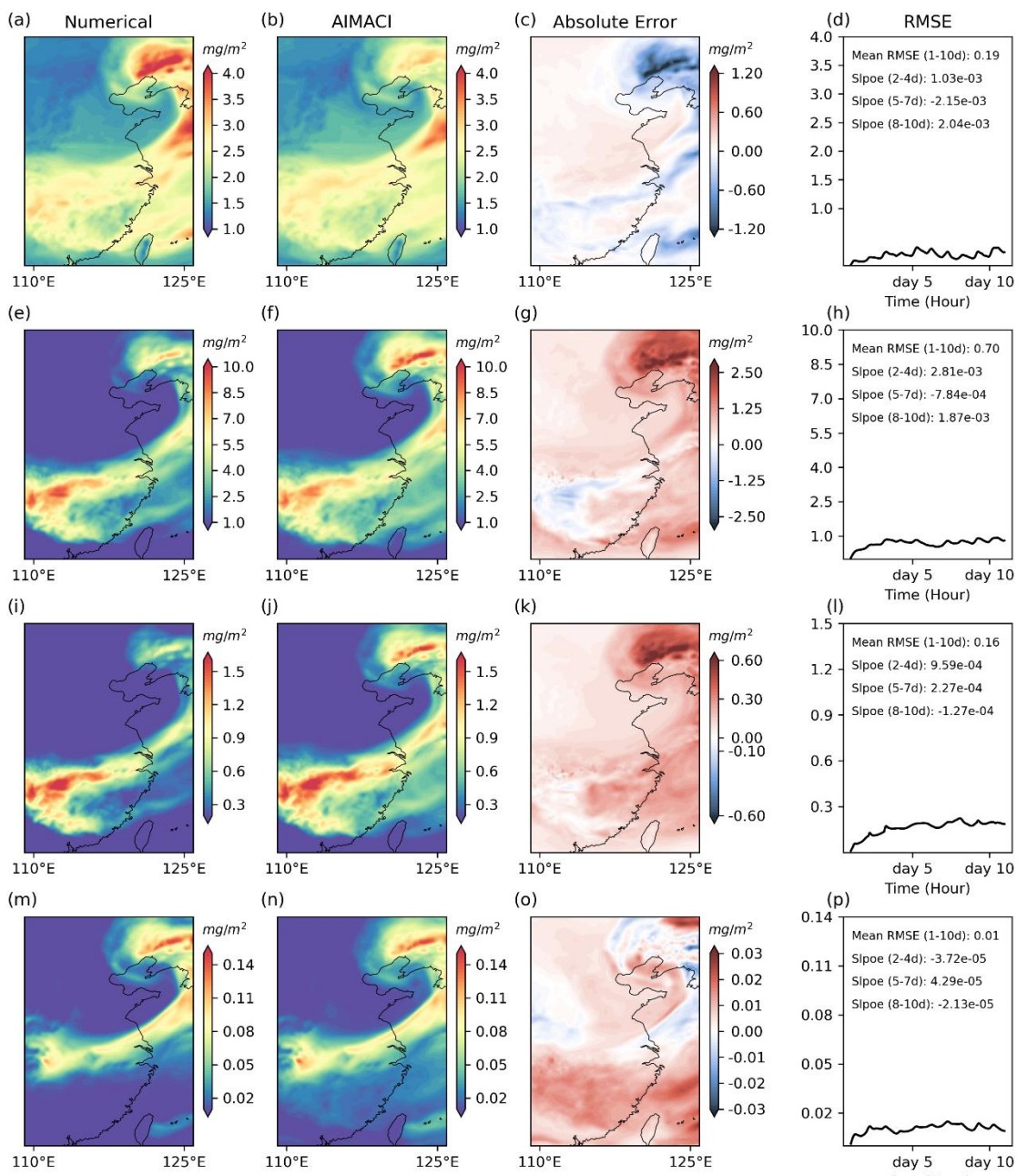

**Figure 3:** Sulfate column concentration simulations across different size bins. The first and second column depict the spatial distribution at the 10-day continuous simulation's end (2019-03-30 00:00 UTC), as simulated online by both the numerical scheme and the AIMACI scheme. The third column is the absolute error between them. The fourth column shows the temporal evolution of the hourly RMSE over the 10-day period. The mean RMSE (unit: mg /m$^2$) for all days and the slope (unit: mg m$^{-2}$ h$^{-1}$) for different simulation stages (2-4day, 5-7day, 8-10day) are given inset.

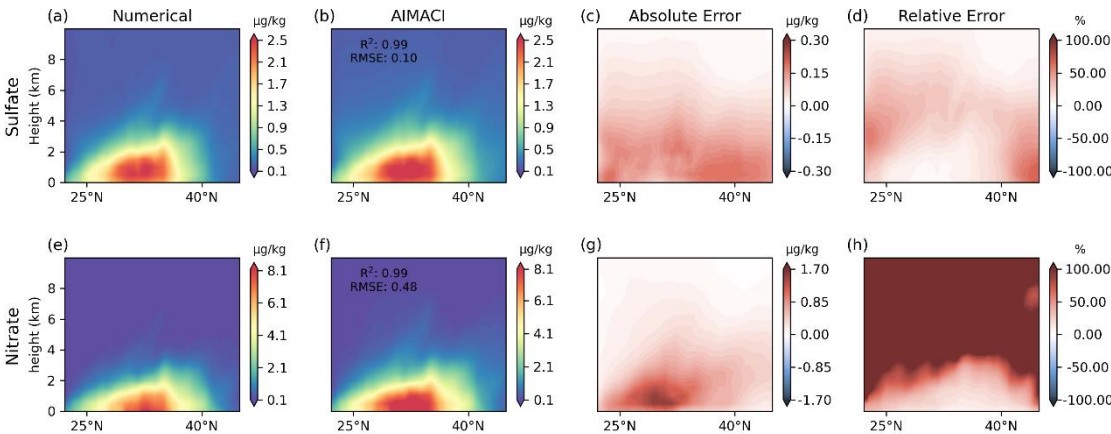

**Figure 4:** Zonal mean total concentrations (summed across 4 size bins) of sulfate and nitrate between 109.1°E and 125.9°E, as simulated online by both the numerical scheme and the AIMACI scheme. Results are averages over the entire 10-day simulation period.


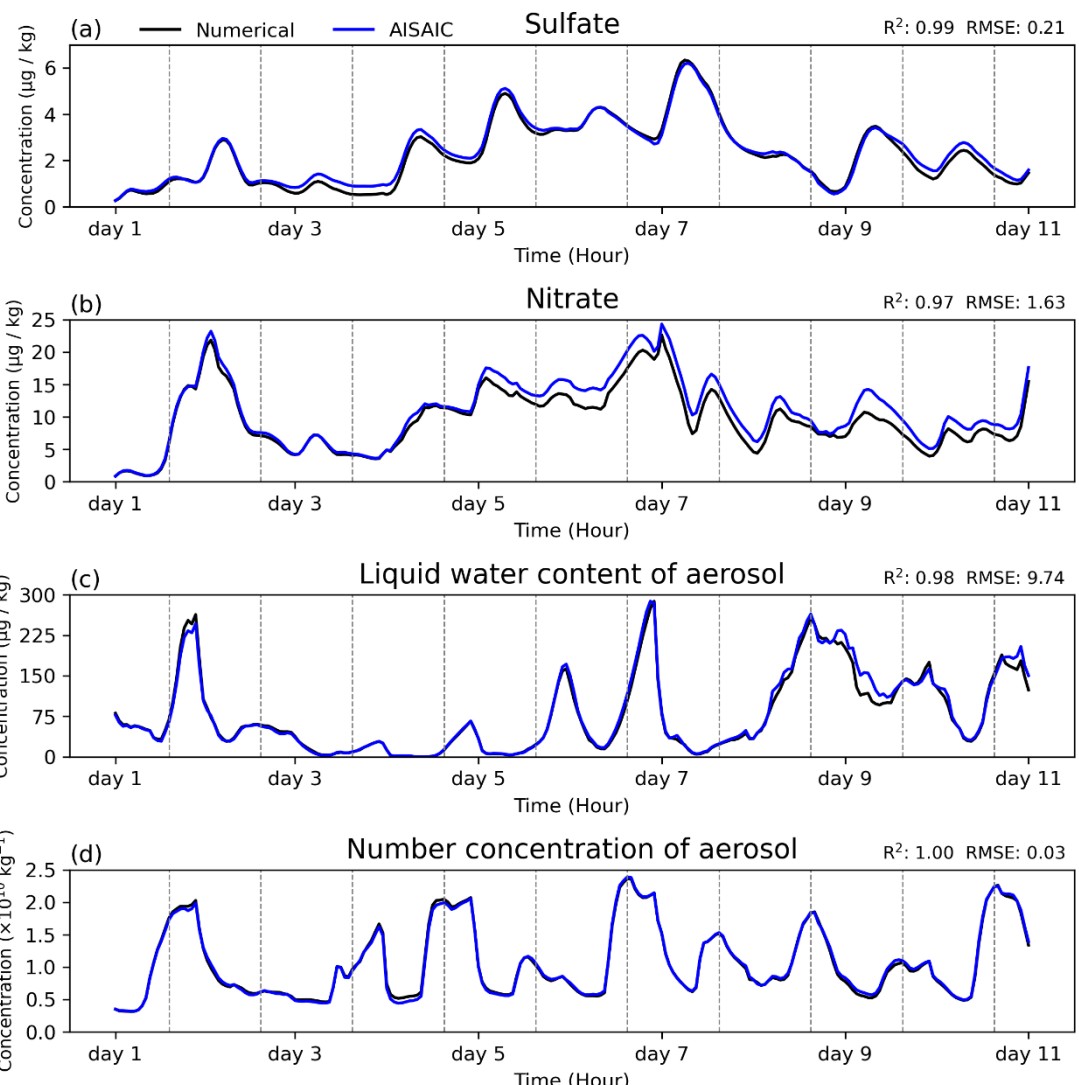


**Figure 5:** Time series of surface total concentrations (summed across 4 size bins) of four key aerosol species (sulfate, nitrate, liquid water content of aerosol, and number concentration of aerosol), as simulated online by both the numerical scheme and the AIMACI scheme. Results represent the calculated averages for the Yangtze River Delta region (119.1°E~121.9°E, 30.1°N~31.9°N). The grey vertical lines mark the time intervals from 0 to 24 hours in Beijing Time for each corresponding

day.

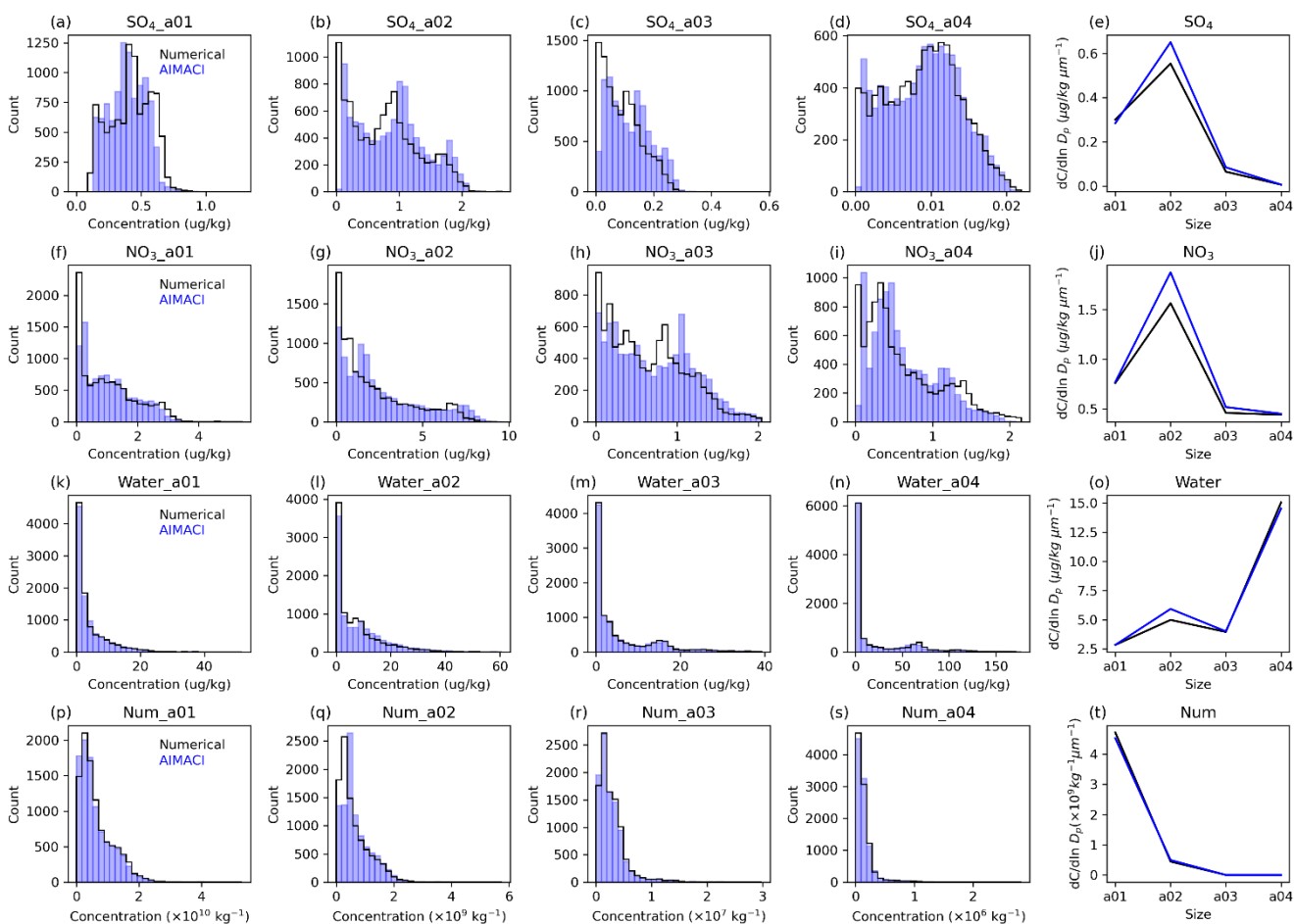

**Figure 6:** Frequency distributions of different particle sizes for the surface concentrations of four key aerosol species (sulfate (SO4), nitrate (NO3), liquid water content of aerosol (Water), and number concentration of aerosol (Num)), as simulated online by both the numerical scheme and the AIMACI scheme. The last column showcases the particle size distributions of these key aerosol species surface concentrations. The results are calculated by covering the region spanning from 109.1°E to 125.9°E and from 22.1°N to 44.9°N, with the data being averaged over the entire simulation period.

### 3.2.2 Robust Generalization Ability

In the preceding section, we have demonstrated the AIMACI scheme's remarkable success in simulating the 3D spatiotemporal distributions and PSD of various aerosol species concentrations. Building upon these findings, this section delves into an exploration of the AIMACI scheme's generalization ability under diverse environmental conditions, a critical aspect for its future integration into climate models to mitigate uncertainties stemming from oversimplified or absent aerosol processes. To evaluate this, we conducted a series of supplementary experiments. These experiments involved a comparative

analysis of one-month simulations for each of the four seasons—spring, summer, autumn, and winter—between the numerical scheme and the AIMACI scheme. This comprehensive evaluation ensures that the AIMACI scheme's performance is not limited to specific conditions but is consistently accurate across a range of environmental scenarios, thereby bolstering its applicability and reliability in climate modeling endeavors.

Figure 7 illustrates the monthly average surface total concentrations of nitrate simulated by the AIMACI scheme across
four seasons, along with the absolute and relative error plots. The results reveal distinct seasonal variations in nitrate surface concentrations, with higher values in January and lower in July. This seasonal contrast is primarily due to increased anthropogenic emissions from winter heating activities, leading to a surge in nitrate aerosols. In contrast, summer months are characterized by meteorological phenomena such as the East Asian monsoon, which enhances atmospheric dispersion and removal of aerosols, resulting in reduced nitrate concentrations at the surface. The nitrate concentrations are mainly
concentrated between the latitudes of 25°N and 40°N, but there are discernible differences in the distribution of high-concentration zones and concentrations over sea areas across different months. Despite being trained on data from only 16 days in March, the AIMACI scheme demonstrates a remarkable ability to reproduce these distribution characteristics under different environmental conditions. This is supported by $R^2$ values larger than 0.93, indicating a strong agreement between the AIMACI scheme and the numerical scheme. Further analysis of the absolute error plots in Figure 7 shows that in January,
when nitrate concentrations are relatively high, the model tends to underestimate the high-value regions, whereas in months with lower concentrations, the model generally overestimate the nitrate concentrations. For the low concentration areas, the absolute error plot reveals a slight overestimation across all months. However, due to the small values involved, the relative error is more prominent in these regions compared to areas with higher concentrations.

Figure 8 presents the time series of surface total concentrations of nitrate across different environmental conditions for
the four seasons, as simulated by the AIMACI scheme for the Yangtze River Delta region. The analysis reveals that January exhibits more pronounced fluctuations in nitrate surface concentrations, with peaks surpassing 30 µg/kg, while other months display more rapid variations. Reproducing hourly concentration changes is more challenging than simulating multi-day average concentrations, as the latter can offset positive and negative errors. Nevertheless, the AIMACI scheme maintains high consistency with the numerical scheme, with $R^2$ values no lower than 0.86, demonstrating its robust generalization
capabilities. Although there are some biases, they do not accumulate over time but stabilize within a range of 5 µg/kg, fluctuating and even decreasing to 0. This is because, in online continuous simulations, aerosol concentrations are controlled not only by the ACI processes simulated by AIMACI but also altered by other processes in the numerical model such as emission, dry deposition, wet scavenging, and transport. In offline simulations, aerosol concentrations are solely controlled by the ACI process, so multi-step autoregressive simulations may show significantly different bias trends.

Additionally, we note that the simulation errors in July are notably higher than in other months. Considering that July is a typhoon-prone season, while our training data only includes partial data from March, we hypothesize that the lack of representation of such distinct seasonal weather events in our training data may contribute to the larger bias in July. To test this hypothesis, we plotted the minimum values of the Mean Sea Level Pressure (MSLP) in the training data and simulation

results of different months, as shown in Figure 9. The time series of the minimum MSLP for different months exhibit some

differences from the training data, with July showing the most significant difference. This may be one of the contributing factors leading to the overall suboptimal simulation performance in July. Moreover, we observed three lowest MSLP in the time series for April, July, and October that are likely corresponding to typhoon events, as indicated by the red points in the figure. Since the lowest MSLP in October is at the start of the simulation, where the concentration of pollutants is almost zero, it may not be diagnostic of simulation bias. Therefore, we additionally selected a relatively later time point, combined

with the moments corresponding to the three lowest values, and plotted the spatial distribution of nitrate surface concentrations at these four moments, along with corresponding MSLP and wind fields, as shown in Figure 10. These results reveal that the lowest values in both July and October correspond to typhoons over the western Pacific, while the lowest value in April corresponds to a low-pressure system over land. The absolute error plot in Figure 10 shows that in the regions affected by typhoons, the simulation errors are significantly increased (Figures 10h, p), whereas the accuracy of the AIMACI

scheme in simulating the low-pressure system in April is not significantly impacted. This indicates that the AIMACI scheme the AIMACI scheme possesses a certain degree of generalization capability, but it still has limitations when facing significantly different seasonal weather events, such as typhoons. To surmount these limitations and bolster the precision of the AIMACI scheme, future iterations should consider integrating a more comprehensive and diverse training dataset that accounts for a broader spectrum of environmental conditions, notably incorporating additional seasonal meteorological

phenomena such as typhoons. The augmentation of the training dataset would facilitate the fine-tuning of the AIMACI scheme, thereby enhancing its reliability in delivering robust simulation outcomes.

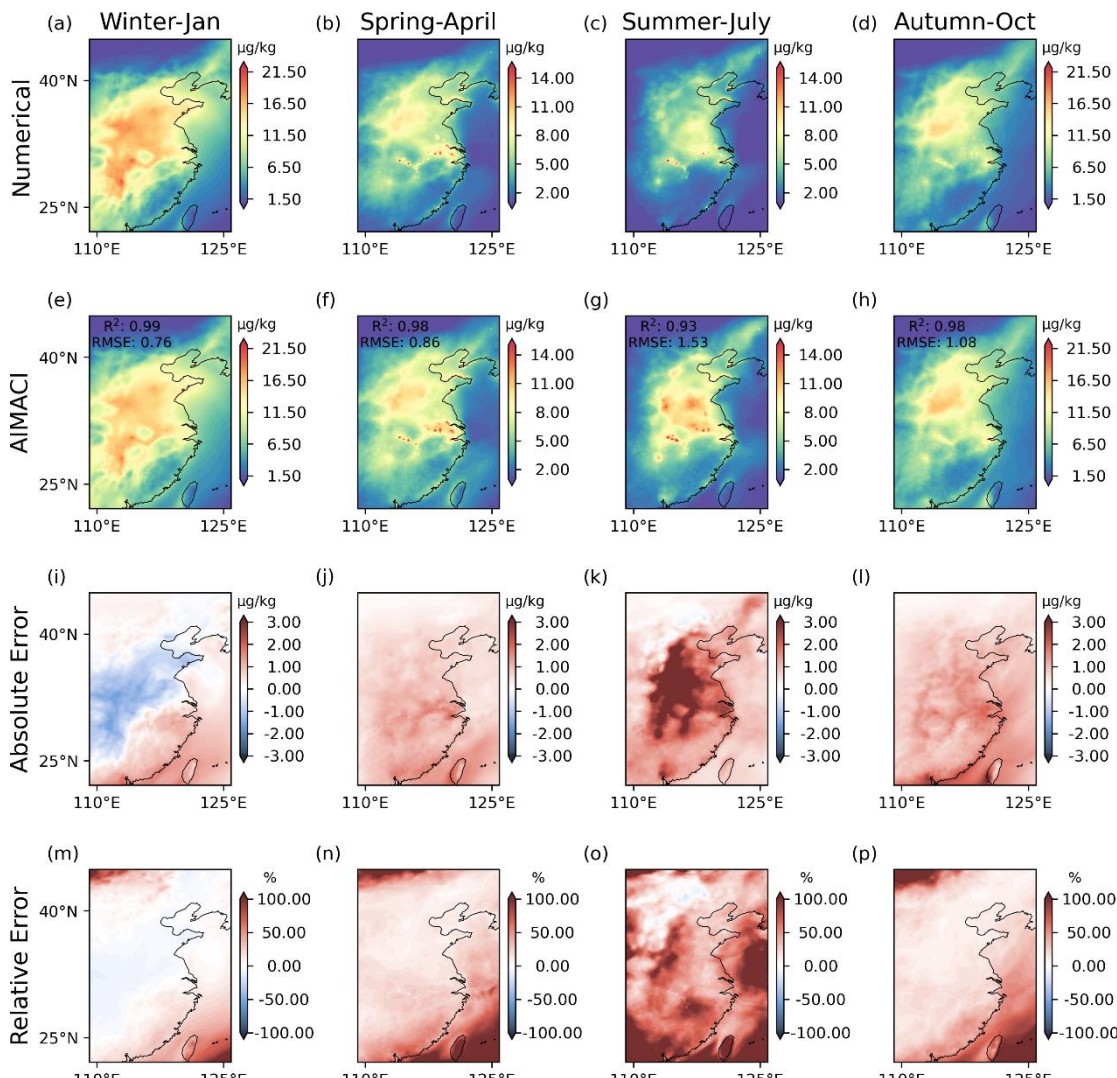

**Figure 7:** Monthly average surface total concentrations (summed across 4 size bins) of nitrate for different environmental conditions across seasons, as simulated online by both the numerical scheme and the AIMACI scheme.


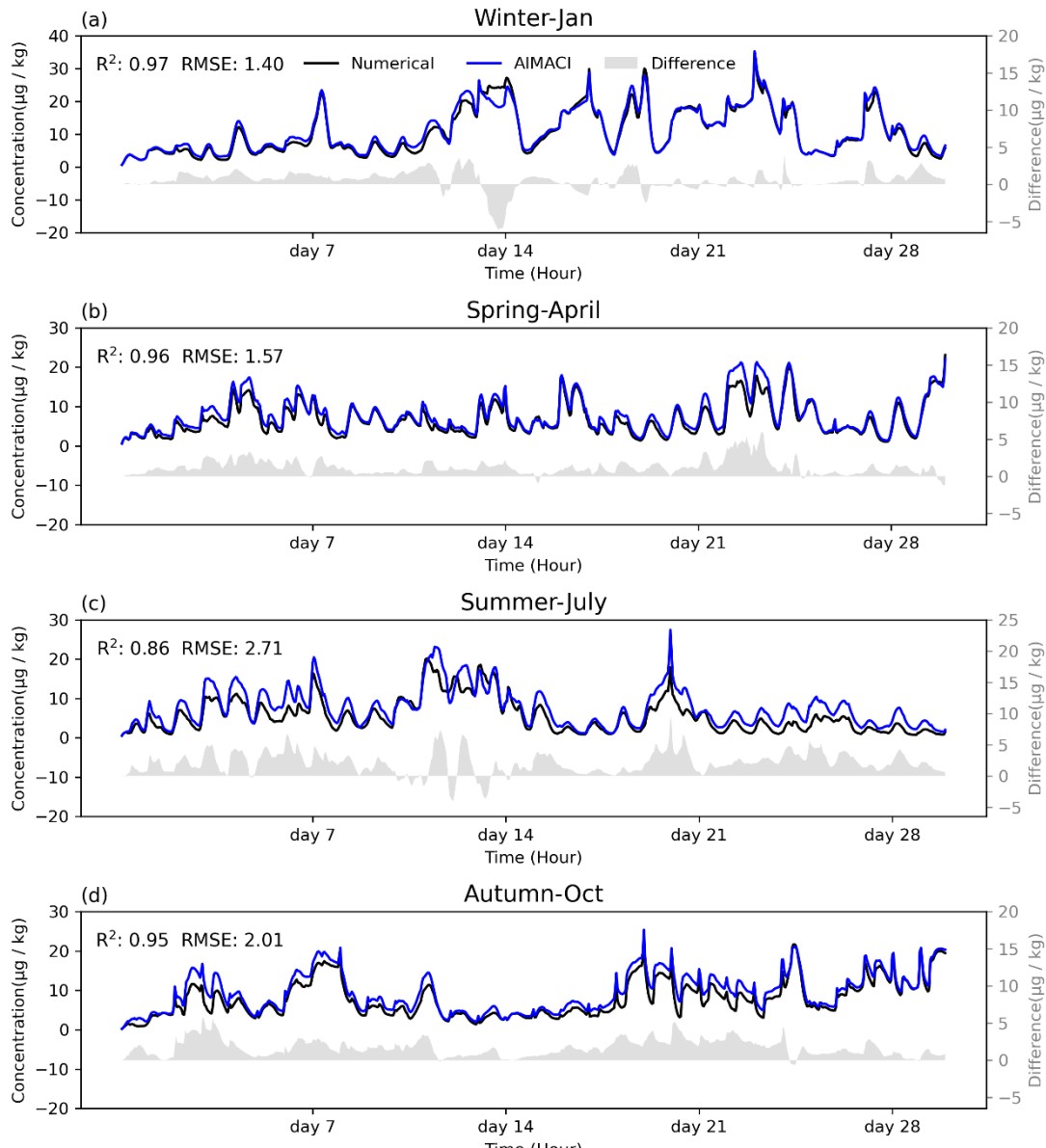

**Figure 8:** Time series of surface total concentrations (summed across 4 size bins) of nitrate for different environmental conditions across seasons, as simulated online by both the numerical scheme and the AIMACI scheme. The grey bar is the difference between the two schemes. Results represent the calculated averages for the Yangtze River Delta region (119.1°E~121.9°E, 30.1°N~31.9°N).

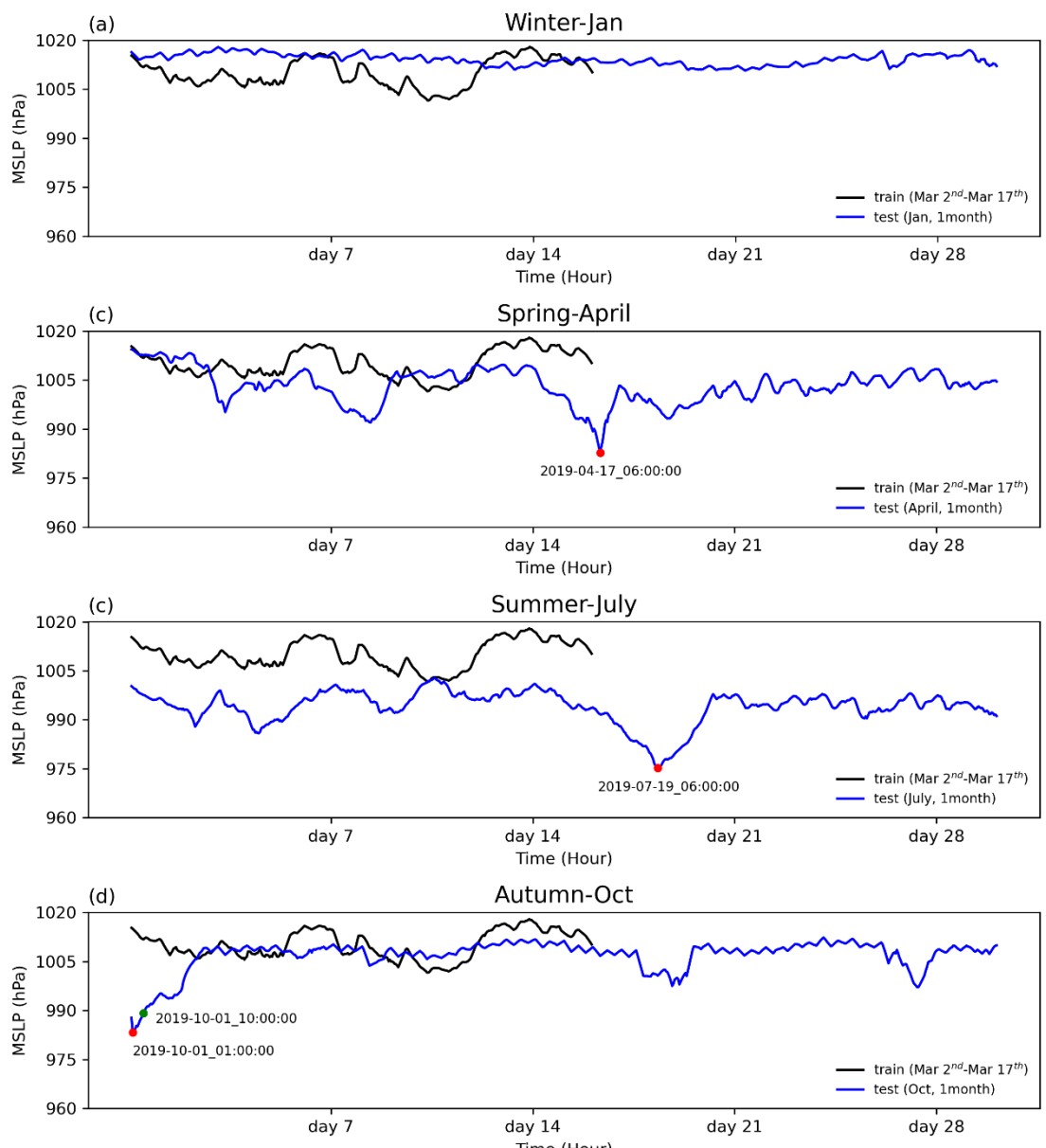

**Figure 9:** Time series of the minimum of Mean Sea Level Pressure (MSLP) in the training data and simulation results of different environmental conditions across seasons. The red points represent the lowest MSLP in the corresponding time series and the date of its occurrence. Since the lowest MSLP in October is at the start of the simulation, where the concentration of pollutants is almost zero, it may not be diagnostic of simulation bias. Therefore, we additionally selected a relatively later time point, as shown by the green point.

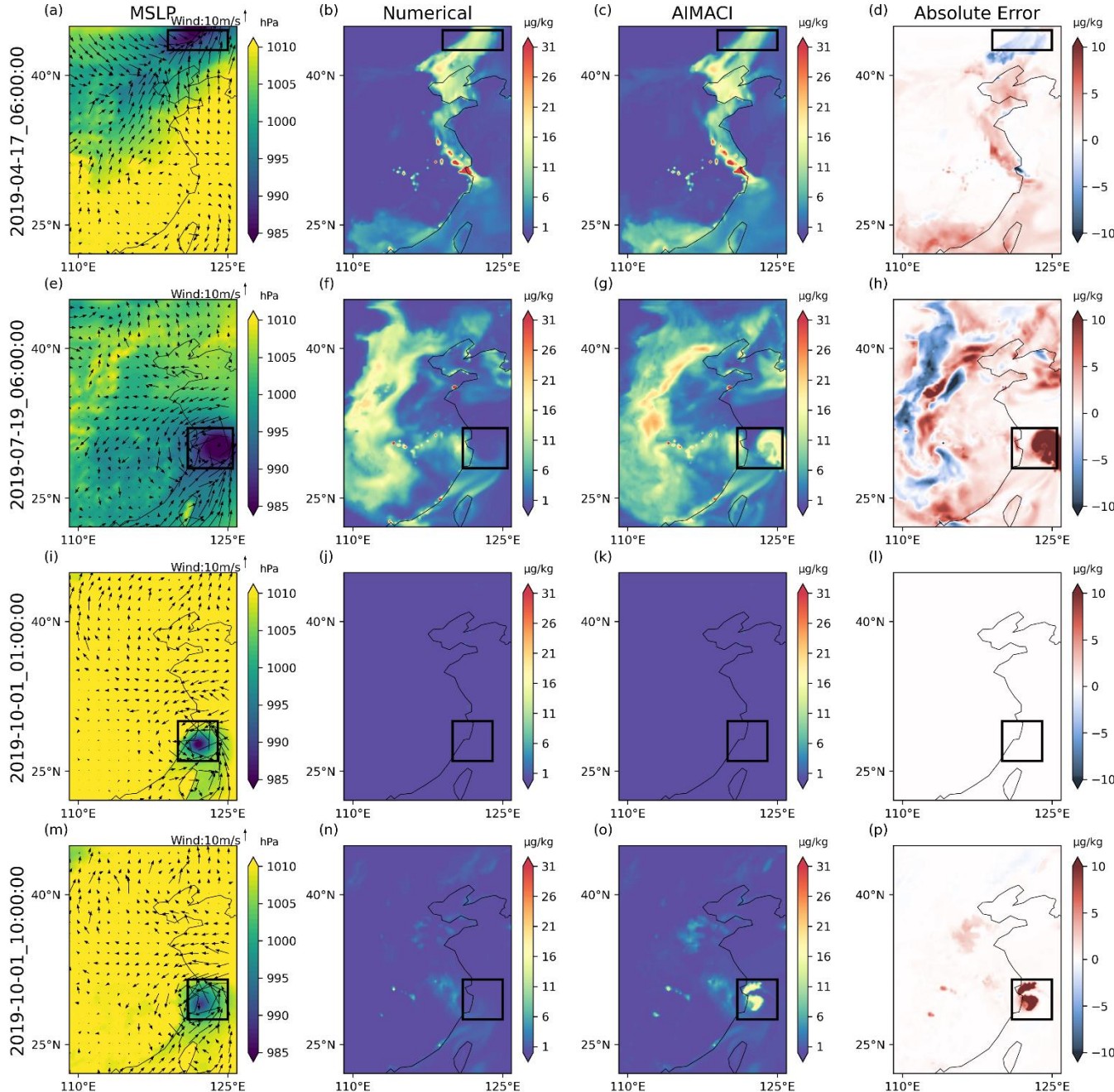

**Figure 10:** The spatial distribution of nitrate surface concentrations at four specified time points, along with corresponding Mean Sea Level Pressure (MSLP) and wind fields. The black box corresponds to the area with the lowest MSLP.

### 3.2.3 High Computational Efficiency

A primary motivation for the development of the AIMACI scheme is the potential for increased computational
efficiency offered by AI schemes compared to conventional numerical schemes. However, past research has indicated that
such computational efficiency gains are not always guaranteed (Keller and Evans, 2019), necessitating a direct comparison
of the computational speeds of the AIMACI scheme and the numerical scheme. Given that the WRF-Chem, written in
Fortran, is not conducive to GPU acceleration, we conducted offline tests of the AIMACI scheme's computational speed on a
GPU and compared it with the numerical scheme on a CPU, where the AIMACI scheme was coupled into the WRF-Chem.
Figure 9 demonstrates that when utilizing a single CPU core, the AIMACI scheme achieves a computational speedup of
approximately 5×, with a time cost of 48.51 seconds compared to the numerical scheme's 229.74 seconds. This advancement
is further amplified when employing a single GPU, the AIMACI scheme completes the computation in a mere 0.83 seconds,
which is approximately 277 × faster than the numerical scheme running on a single CPU core. Although we have not yet
tested the online simulation speed of the physics-AI hybrid model on a GPU, it is reasonable to anticipate that future
implementation of heterogeneous computing platforms, integrating both CPUs and GPUs, will yield significant
enhancements in computational efficiency.

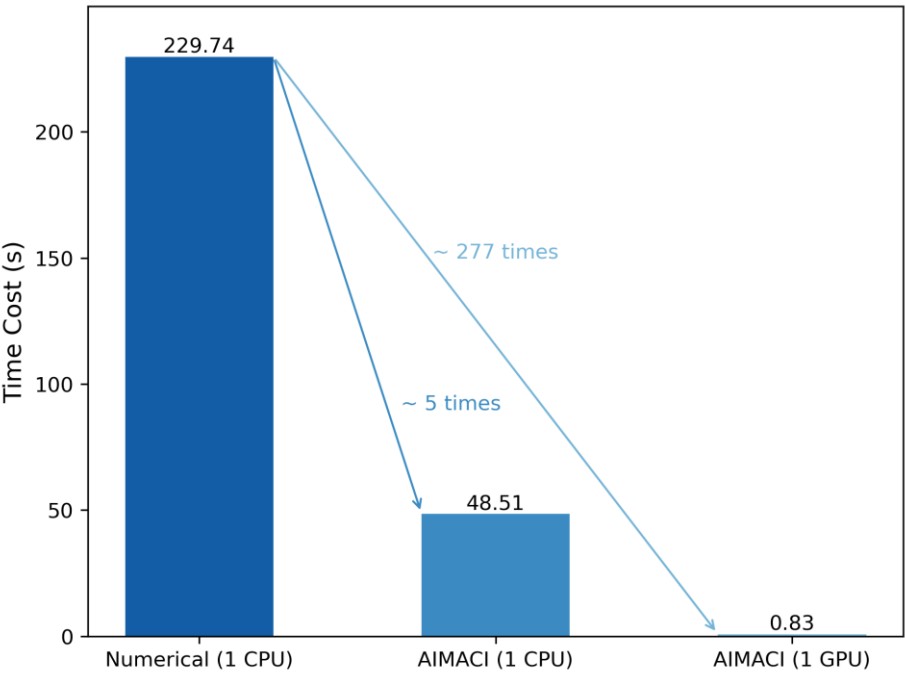

**Figure 11:** Comparison of computational speeds between the numerical scheme and the AIMACI scheme under different
computational configurations. The time cost for the GPU is measured in a mode where the AIMACI scheme is not yet

coupled to the model, while the time cost for the CPU is measured in a mode with the AIMACI scheme coupled into the model. The calculations are based on simulating the concentrations of 37 chemical species across 720,300 grid cells.

**4 Conclusions**

This study develops and evaluates a novel Artificial Intelligence Model for Aerosol Chemistry and Interactions, termed AIMACI, with a special focus on addressing the long-standing challenge of significant computational burden associated with enabling numerical scheme for simulating ACI in atmospheric models. The differential equations governing aerosol chemistry are notably stiff, coupled with a stringent time integration scheme required for numerical stability, resulting in limited breakthroughs in simulation speed with available numerical techniques. While previous studies have explored AI schemes as alternatives for conventional numerical schemes for simulating photochemical processes in atmospheric models, the use of AI schemes for simulating aerosol chemistry and interactions (ACI) has not been studied. Moreover, ACI processes encompass not only chemical reactions but also a suite of highly nonlinear processes, including chemical reactions and phase equilibrium, gas-particle partitioning, particle size growth, coagulation and nucleation. This study, therefore, aim to addresses the critical question of whether an AI scheme can effectively supplant the entire numerical scheme for these processes, achieving both high fidelity in simulation accuracy and a marked increase in computational efficiency.

To this end, the AIMACI scheme was established based on the state-of-the-art Multi-Head Self-Attention algorithm, renowned for its powerful nonlinear representation capabilities. This algorithm can efficiently capture complex reaction relationships between different chemical species, remaining robust even as the number of simulated species increases. In an offline mode, where the AIMACI scheme was not yet integrated with a 3D numerical model, it demonstrated remarkable statistical metrics on a test dataset, all 61 evaluated species exhibiting an average $R^2$ of 0.98, and an average NMB of 3.02%. This high degree of consistency between the numerical and AIMACI schemes lays a solid foundation for further online continuous simulations.

To facilitate the coupling of the Python-written AIMACI scheme with Fortran-based numerical models, we utilized PyTorch's TorchScript and LibTorch tools to encapsulate the AIMACI scheme into a static library callable by the numerical model. This approach entails minimal changes to the existing numerical model's codebase and offers a highly flexible and easily plug-and-play solution for coupling AI algorithms of diverse complexities with a range of numerical models.

Employing physics-AI hybrid model, we implemented additional experiments to evaluate the online simulation performance of the AIMACI scheme. The 10-day continuous simulation results indicate that the AIMACI scheme not only accurately captures the spatiotemporal distribution of various aerosol species but also effectively reproduces their size distributions, maintaining stability throughout the simulation period without rapid error growth. The analysis of the RMSE time series for 10-day simulation revealed that the simulation errors do not necessarily accumulate over time. And, the trends in simulation errors for different species may not align at the same time, which could be related to the fact that, during online simulations, aerosol concentrations are influenced not only by ACI but also by other processes within the numerical model. Furthermore, the AIMACI scheme exhibits robust generalization capabilities, applicable across various environmental conditions in all four seasons for month-long continuous simulations, despite being trained on data from only 16 days in

March. The simulation results for nitrate's monthly average surface total concentrations and hourly time series illustrate a high degree of consistency with the numerical scheme. However, due to the relative scarcity of high values in the training data and the use of RMSE as the loss function for model optimization, the AIMACI scheme exhibits a common issue found in AI models: underestimating high values and overestimating low values. Additionally, compared to other months, the performance of the AIMACI scheme in July was less accurate. Our analysis indicates that this is primarily due to the significant difference between the meteorological conditions in July and those in the training data. Seasonal weather events, such as typhoons, have a notable impact on the performance of AIMACI. Therefore, in future iterations of AIMACI, we plan to use a more diverse training dataset, particularly one that includes seasonal weather events, to further enhance its performance.

In terms of computational speed, the AIMACI scheme is approximately 5 times faster than the conventional numerical scheme when predicting 720,300 grid points with 61 chemical species using a single CPU core. This speedup increases significantly to about 277 times faster when utilizing a GPU. Future simulations on heterogeneous platforms, integrating both CPUs and GPUs, are expected to further improve this speedup ratio. This anticipated enhancement will enable higher spatial resolutions, extended simulation durations, and substantially reduced computation times for chemical transport models.

An important outcome of this work is the first-time successful application of an AI scheme to replace entire numerical scheme for ACI within the numerical model, achieving fast, accurate, and stable month-scale simulations. The high fidelity in reproducing the complex spatiotemporal distributions and PSD of aerosol species including water content in aerosols, coupled with a significant acceleration, highlights the potential of the AIMACI scheme in advancing climate modeling and atmospheric science. As the first step, we concentrate on inorganic aerosols, which are a fundamental aspect of atmospheric chemistry. While organic aerosols also play a crucial role, the inherent chemical complexity and the absence of a convincing numerical scheme for AI scheme to emulate have led us to defer their inclusion. However, the exploration of AI scheme for organic aerosols is firmly on our agenda for future studies, with the aim of integrating them into our comprehensive AIMACI scheme. At the same time, based on the convenience and flexibility of the method we employed in this study— coupling the AIMACI scheme with the numerical model via a static library—as well as the robust generalization ability demonstrated by the AIMACI scheme, we are actively testing its application in global long-term simulations. This not only helps to reduce uncertainties in global models due to simplified or absent detailed ACI but also advances our comprehension of aerosol behavior, vital for climate studies and policy-making. By harnessing the power of artificial intelligence in conjunction with sophisticated atmospheric models, we aspire to achieve faster, more accurate, and reliable predictions of future climate scenarios.

## Code availability

The hybrid version of WRF-Chem with both physics and AI schemes is publicly accessible. The source code for this hybrid model, along with the parameter file for the Multi-Head Self-Attention model used in the Artificial Intelligence Model for Aerosol chemistry and Interactions (AIMACI) is available under an open license at https://zenodo.org/records/13736859.

## Author contributions

Zihan Xia and Chun Zhao designed the experiments, conducted and analyzed the simulations. All authors contributed to the discussion and final version of the paper.

## Competing interests

The authors declare that they have no conflict of interest.

## Acknowledgments

This research was supported by the National Key Research and Development Program of China (No. 2022YFC3700701), the Strategic Priority Research Program of Chinese Academy of Sciences (XDB0500303), National Natural Science Foundation of China (41775146), the USTC Research Funds of the Double First-Class Initiative (YD2080002007, KY2080000114), the Science and Technology Innovation Project of Laoshan Laboratory (LSKJ202300305), the National Key Scientific and Technological Infrastructure project "Earth System Numerical Simulation Facility" (EarthLab), and CMA-USTC Laboratory of Fengyun Remote Sensing. The study used the computing resources from the Supercomputing Center of the University of Science and Technology of China (USTC) and the Qingdao Supercomputing and Big Data Center.

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
