# Peer review of "Toward a Learnable Artificial Intelligence Model for Aerosol Chemistry and Interactions (AIMACI) based on the Multi-Head Self-Attention Algorithm"

_EGUsphere, 2024_

## Author Comment (AC1)

**Dear Editors:**

We sincerely thank you for facilitating the review process and for providing us with valuable feedback. The comments we received have been instrumental in enhancing the quality and clarity of our manuscript.

We have carefully considered all the comments and have made comprehensive revisions to address each point raised. Our detailed responses to the reviewers' comments are provided below.

Yours sincerely,
Zihan Xia and co-authors

**Reviewer #1**

*General comments:*

- *Xia et al. apply a multi-head self-attention (MHSA) transformer machine-learning model for aerosol chemistry for the first time in the WRF-Chem CTM. The results are promising and fascinating. However, many issues with language and the presentation of the results need refinement and tempering.*

**Response:** We thank you for your time and expertise. Your constructive criticism has been instrumental in refining our work. We have carefully addressed each of your specific comments, as detailed in our point-by-point responses below. We believe these revisions have enhanced the manuscript and hope that they adequately address your concerns.

*Specific comments:*

- *1. Abstract: I would not use the descriptor "Remarkably" if these are similar speedups we are seeing in the literature for CPU and GPU speedups of ML chemical solvers.*

**Response:** We appreciate your observation regarding the use of the term "Remarkably" and agree that it may not be an appropriate choice given the context of similar speedups reported in the literature for machine learning chemical solvers. Upon your suggestion, we have elected to replace "Remarkably" with a more suitable connector that maintains the flow of the sentence while accurately reflecting the significance of our findings. We believe that the term "Notably" provides a balanced alternative, emphasizing the speedup achievements without overstatement. The updated text now reads: "Notably, it exhibits a ~5× speedup with a single CPU and ~277× speedup with a single GPU compared to conventional scheme."

- *2. Abstract: "While global long-term simulations have not yet been implemented, AIMACI's robust generalization capability, coupled with our easily plug-and play solution, paves the way for its coupling into global climate models for further testing in near future. This advancement promises to enhance the precision and efficiency of atmospheric aerosol simulations in climate modeling.". -I would temper this assessment. The results do indeed seem promising but you have not shown stable, global, year-long simulations using this fully ML-learned replacement and past studies have shown that this is not guaranteed at all even if shorter-term simulations work.*

**Response:** Thank you for your insightful comments. We appreciate your caution regarding the assertion of AIMACI's potential in global climate models without the demonstration of stable, year-long simulations. You are correct that the absence of such long-term validation is a significant limitation. It is for this reason that we are actively working to couple both AI-based chemical schemes—photochemistry and aerosol chemistry—within the atmospheric physical-chemical fully coupled global variable-resolution atmospheric numerical model iAMAS, developed by our research group. This integration will allow us to conduct a more comprehensive and nuanced evaluation of AIMACI's performance. We fully agree that definitive claims about

AIMACI's impact on climate modeling should be reserved until it has undergone rigorous testing over longer timeframes. To this end, we are committed to carrying out these tests and will share the results in our future publications. In light of your feedback, we have revised the manuscript to include a more nuanced statement that acknowledges the current limitations and the need for further testing. The updated abstract now reads: "However, the stability of AIMACI for year-scale global simulations remain to be seen, requiring further testing. AIMACI's generalization capability and its plug-and-play nature suggest potential for future coupling into global climate models, which are expected to enhance the precision and efficiency of aerosol simulations in climate modeling that neglects or simplifies ACI processes."

- *3. Introduction: How do aerosol schemes compare with gas-phase chemical mechanisms in terms of computational cost? Aren't many schemes dealing with heterogeneous chemistry separated from the gas-phase mechanism and incur lower computational overhead? Perhaps slightly more information/discussion is needed to contextualize the cost of the aerosol scheme compared to other components of chemistry/climate models. The background on the previous AI models is great, however.*

**Response:** We are grateful for your insightful suggestion. Both gas-phase chemical mechanisms and aerosol schemes are computationally demanding components in the chemical module. The specific comparison of computational costs between them depends on the particular schemes chosen, which are mainly influenced by factors such as the number of species and reaction equations included. To address your comments, we have conducted a detailed analysis of the computational time required for the CBMZ gas-phase chemical mechanisms, the 4-bin MOSAIC aerosol scheme focusing on major inorganic aerosols, and other parts of the chemical module within our study. This analysis is now presented as Figure S1 in the manuscript. As shown in Figure S1, the aerosol scheme in our study accounts for 31.4% of the total computational time in the chemical module, while the gas-phase chemical mechanisms account for 9.4%.

We acknowledge that there are indeed schemes dealing with heterogeneous chemistry that are separated from the gas-phase mechanism and incur a lower computational overhead. However, our aim is to replace the full aerosol chemistry and interactions, which include not only chemical reactions and phase equilibrium but also gas-particle partitioning, particle size growth, coagulation, and nucleation. These processes collectively have a significant impact on atmospheric aerosol concentrations and, when considered together, consume the aforementioned 31.4% of the computational time.

In light of this analysis, we have revised the original text as follows: "Aerosol chemistry and interactions (ACI) involve a range of highly nonlinear processes, including chemical reactions and phase equilibrium, gas-particle partitioning, particle size growth, coagulation, and nucleation, which have a significant impact on the concentration of atmospheric aerosols. Numerical models stand as indispensable analytical tools, pivotal for comprehending the aforementioned phenomena, and are instrumental in air quality management and the formulation of mitigation strategies for climate change. However, coupling ACI into these models poses a significant computational challenge (Carmichael et al., 1999; Ebel et al., 2006). As shown in

Figure S1, which displays the proportion of computational time for different parts of the chemistry module, the Model for Simulating Aerosol Interactions and Chemistry (MOSAIC) scheme with just four bins for major inorganic aerosols already accounts for 31.4% of the total computational time."

[Figure]

**Figure S1:** Proportion of computational time for different parts of the chemistry module in this study. The photochemistry is modeled using the CBM-Z (Carbon Bond Mechanism Version Z) scheme, which does not account for gas species related to complex SOA, while aerosol chemistry and interactions are simulated using the Model for Simulating Aerosol Interactions and Chemistry (MOSAIC) scheme, which includes only the main inorganic aerosol species and utilizes a 4-bin configuration.

- *4. Introduction: L54: "methodologies for describe the evolution of PSD", should be "describing".*

**Response:** Thank you for your correction. In the revised manuscript, we have corrected the grammatical error.

- *5. Introduction: "Unlike photochemistry which only involves chemical reactions between species, the full aerosol chemistry and interactions encompasses numerous other intricate processes such as nucleation, coagulation, thermodynamics" --> Aren't aerosol schemes also coupled to the chemical mechanism as well? Or are you discussing only microphysics/thermodynamic calculation of aerosol schemes? It is a little vague so far.*

**Response:** Thank you for your meticulous review and for raising this important point of clarification. In WRF-Chem, photochemistry and aerosol chemistry and interactions are indeed closely related in the chemical mechanism but are computationally distinct. In our study, photochemistry is represented by the CBMZ chemical mechanism, while aerosol chemistry and interactions are handled by the MOSAIC scheme. Within the chemistry module, it first calculates the photochemistry, including those gases that are pertinent to aerosol chemistry and interactions, such as $H_2SO_4$ and $HNO_3$, using the CBMZ mechanism. After that, the MOSAIC scheme is employed to address aerosol chemistry and interactions. The

detailed processes included in aerosol chemistry and interactions are chemical reactions and phase equilibrium, gas-particle partitioning, particle size growth, coagulation, and nucleation, as explained in my response to your previous review comment No. 3.

- *6. Introduction: "These advancements hold great promise for the future of climate modeling, enabling fast, accurate, and stable simulations of aerosol chemistry and interactions, thereby reducing uncertainties stemming from simplified representations of these processes." -- I don't think you can claim that this is stable because the time scales are not long enough for climate-relevant (or even seasonal) time scales. They are stable for 1 month, which is good to see but no guarantee that they are stable beyond this (as you have not shown it).*

**Response:** We appreciate your caution regarding the claim of stability for climate-relevant time scales. You are correct that our current demonstration of stability over a one-month period does not guarantee stability over longer periods. We have taken your feedback into account and will further clarify this point in the revised manuscript to ensure a more measured and accurate representation of our findings. Additionally, another reviewer suggested that results should not be included in the introduction. In response, we have removed the specific results and instead emphasized the validation of AIMACI's accuracy, stability, and computational efficiency. The updated paragraph now reads: " To bridge this gap, in this study, we have developed a novel Artificial Intelligence Model for Aerosol Chemistry and Interactions, termed AIMACI, which is based on the Multi-Head Self-Attention algorithm and can been online coupled with a 3D numerical atmospheric model. As the first step, this study focuses on inorganic aerosols, because the chemistry of organic aerosols (i.e., secondary organic aerosols) still has large uncertainties and lacks a convincing numerical scheme for AI scheme to emulate, which certainly deserves further investigation in future. To validate the accuracy, stability, and computational efficiency of the AIMACI scheme, we conducted a series of experiments for both offline simulations (where AIMACI scheme was not coupled to a numerical model) and online simulations (where AIMACI scheme was coupled to a numerical model). The structure of this paper is organized as follows: Section 2 provides a detailed description of the Weather Research and Forecasting with Chemistry (WRF-Chem) model and the establishment of the AIMACI scheme. Section 3 discusses the results, and Section 4 presents the conclusion, outlining the implications of our findings for the field."

- *7. Methods: "Therefore, here we innovate by pioneering a novel Artificial Intelligence Model for Aerosol Chemistry and Interactions (AIMACI), leveraging the MHSA algorithm. The MHSA algorithm, serving as the foundational architecture of state-of-the-art transformer models, has been successfully…". Please tone down this language, it reads as very pretentious. Multi-head attention transformers have existed in the CS/NLP literature for a while. You can claim this is new for aerosol chemistry.*

**Response:** We appreciate your feedback on the tone and phrasing of our manuscript. We have endeavored to strike a balance

between accentuating the novelty of our approach and recognizing the substantial work that has been done in related fields. The revised paragraph is presented below: "Therefore, in this study, we introduce an innovative application of MHSA algorithm in the field of ACI simulation through the development of the Artificial Intelligence Model for Aerosol Chemistry and Interactions (AIMACI). Although the MHSA algorithm has been instrumental in the advancement of state-of-the-art transformer models in domains such as Natural Language Processing, Computer Vision and Weather Forecast, as evidenced by the seminal works of Vaswani et al. (2017), Liu et al. (2021), and Bi et al. (2023), its utilization in ACI simulation represents a new horizon. The algorithm's ability to globally attend to input variables and conduct parallel computations across multiple heads is pivotal in tackling the challenges posed by the curse of dimensionality, capturing complex interdependencies, and significantly enhancing computational efficiency."

● *8. Methods: There does not seem to be much information on how you trained the MHSA transformer either. Did you fine-tune on a pre-trained model? Did you build the MHSA from scratch or use an out-of-the-box GPT model? Or was this completely borrowed and reworked from the other Xia et al paper? Either way, more information is needed on the model development information, and how the training was done (what resources/wall time it took). E.g., what does the tokenization of the WRF-Chem data look like? How were these patches determined/designed, etc?*

**Response:** Thank you for your valuable feedback. We acknowledge that it is important to provide a detailed information of our model development and training methodology in the manuscript. Now we have rewritten Section 2.2.2 (Training and Testing Procedure) that elaborates on the training process and other relevant details. Here is the updated information:

"To generate the training, validation, and test datasets, we conducted the WRF-Chem simulations over East China, spanning the period from 2019-03-01 00:00 UTC to 2019-03-19 23:00 UTC. The simulation result was segmented as follows: the initial 16 days from 2019-03-02 00:00 UTC, were designated as the training dataset, the penultimate day served as the validation dataset, and the final day constituted the test dataset. The simulation was configured with a 0.2° horizontal resolution, covering $140 \times 105$ grid cells within the geographical bounds of 107.1° E to 127.9° E and 19.7° N to 47.5° N, and featured 49 vertical layers extending up to 50 hPa. A dynamic time step of 2 minutes and a chemical time step of 1 hour were employed. For emission and meteorological field, we used the Multi-resolution Emission Inventory for China (MEIC) at 0.25° x 0.25° resolution for 2019 (Li et al., 2017a; Li et al., 2017b), and the NCEP final reanalysis (FNL) data with a 1° x 1° resolution and 6-hour temporal resolution within the simulation domain. Concentrations of aerosol and gas species pertinent to gas-particle partitioning were recorded hourly, along with key meteorological variables influencing chemistry: temperature, pressure, air density, and water vapor mixing ratio. A comprehensive list of variables used for training the AIMACI scheme is presented in Table 1. Due to computational cost considerations, the period of the training dataset is not extensive; however, the volume of training samples is large due to the hourly chemical time step and the fine spatial resolution of our simulation. With 140 by 105 grid cells, 49 vertical layers, and 24 hours in a day, the total number of training samples amounts to 276,595,200

(140x105x49x24x16), reaching the hundred million scale. This large dataset provides a rich and diverse set of samples for training, ensuring that the AI model does not suffer from a lack of convergence due to insufficient data.

In the training of the AI model we built from scratch, each training sample included 65 input features (4 meteorological variables, 5 gas species, and 14 aerosol species with 4 size bins) and 61 output targets (5 gas species and 14 aerosol species with 4 size bins). All features and targets underwent min-max normalization to standardize the data. We employed the PyTorch deep learning framework for model training, with a batch size of 2048, an initial learning rate of 0.001, and the Adam optimization algorithm. The Mean Squared Error (MSE) was used as the loss function. To optimize the training process, we implemented a learning rate decay strategy using the ReduceLROnPlateau scheduler, along with an early stopping mechanism after 10 consecutive epochs without improvement in the validation loss. All other hyperparameters not mentioned are kept at their default values. For this study, we trained the model using three GPUs for approximately three days, and the model achieved optimal performance at epoch 32."

- *9. Methods: I don't find Figure 1 very informative or helpful. Doesn't situate where the ML model is within WRF-Chem, does not give a sense of the dimensionality either. Table 1 is more useful than this, so please rethink this use of space.*

**Response:** Thank you for your valuable feedback. We have revised Figure 1 to make it more informative and clearer. The updated version explicitly highlights the structure of the AIMACI model, the coupling method, and where AIMACI couples within the WRF-Chem. Additionally, it now details the aerosol chemistry and interactions processes replacing by AIMACI. Accordingly, we also revised the description of Figure 1 as follows: "Figure 1 illustrates the integration of AIMACI scheme in our hybrid atmospheric model with physics and AI schemes (physics-AI hybrid model), and also provides a schematic representation of the AI model architecture that is utilized within the AIMACI scheme."

[Figure]

**Figure 1:** The Artificial Intelligence Model for Aerosol Chemistry and Interactions (AIMACI) in the Weather Research and Forecasting with Chemistry (WRF-Chem). The trained AIMACI is packaged into a static/dynamic library using TorchScript and Libtorch, and can be called by WRF-Chem through an interface to replace the aerosol chemistry and interactions numerical scheme, while the remaining processes maintain the original numerical scheme.

- *10. Methods: "After training, the AIMACI scheme was flexibly coupled into WRF-Chem, utilizing TorchScript and Libtorch tools officially provided by PyTorch. This coupling approach encapsulates the AIMACI scheme within a static library, minimizing alterations to the original codebase while offering a lightweight, adaptable, and easily plug-and-play solution". How does this approach compare to the other common approach of using C Foreign Function Interface (CFFI) to create C-style bindings for Python scripts? The latter also seems flexible and lightweight without altering the code base. So if there is an inherent advantage of the TorchScript approach that should be stated.*

**Response:** Thank you for your comment. CFFI stands for C Foreign Function Interface for Python, and its core goal is to call C code from Python without the need to learn a third language. It is not inherently designed for calling Fortran to invoke AI models. In contrast, the TorchScript and Libtorch tools are specifically designed by PyTorch to facilitate the execution of AI models within C++ environments, offering several advantages over CFFI: (1) Performance: TorchScript and Libtorch are part of the PyTorch ecosystem, which has been optimized for performance. This official support ensures that the AI models run efficiently in C++ compared to third-party libraries such as CFFI. (2) Compatibility: LibTorch supports cross-platform deployment, providing more flexibility compared to CFFI, which often requires platform-specific considerations for compatibility. (3) Ease of Use: The official PyTorch documentation provides comprehensive examples on how to package PyTorch-trained AI models into static or dynamic libraries using TorchScript and Libtorch. This significantly simplifies the

process, as we mainly need to write an interface to call the AI model, contrasting with CFFI, which would necessitate writing the entire codebase from scratch. We have revised our manuscript to highlight these points as follows: "After training, the AIMACI scheme was packaged into a static library and then flexibly coupled into WRF-Chem, utilizing TorchScript and Libtorch tools officially provided by PyTorch. Compared to using the third-party libraries like CFFI (C Foreign Function Interface for Python), our coupling approach offers several advantages. Firstly, from a performance perspective, TorchScript and Libtorch tools are part of the PyTorch ecosystem, optimized for running AI models in C++ environments, thus providing faster execution than third-party libraries. Secondly, in terms of compatibility, LibTorch supports cross-platform deployment, offering more flexibility than third-party libraries. Lastly, regarding ease of use, the official PyTorch documentation provides comprehensive examples on how to package PyTorch-trained AI models into static or dynamic libraries using TorchScript and Libtorch. This significantly simplifies the process, as we mainly need to write an interface to call the AI model, contrasting with third-party libraries, which would necessitate writing the entire codebase from scratch. Therefore, this coupling approach minimizes alterations to the original codebase and offers a lightweight, adaptable, and easily plug-and-play solution. It is capable of encapsulating a wide range of complex AI algorithms and coupling them with diverse atmospheric and climate models."

- ***11. Methods: "the initial 16 days from 2019-03-02 00:00 UTC, were designated as the training set, the penultimate day served as the validation set, and the final day constituted the test set." Does the training or the results change if these days are random? This seems like a very short training time window to draw generalizations for aerosol chemistry across an entire domain. Is this long enough to cycle through all the species residence times/lifetimes?***

**Response:** Thank you for your valuable comment. We know that the performance of artificial intelligence models largely depends on the quality and quantity of training data. If the amount of training data is large and the samples are diverse, theoretically, the performance of the model will be significantly improved. However, if the entire dataset is static and the proportions allocated to the training, validation, and testing sets are predetermined, then even if the assignment of data to these sets is performed randomly, the impact on the model's performance should be negligible.

The choice of a 16-day training period was made after careful consideration of several critical factors. Firstly, extending the training period would significantly increase both the numerical simulation costs and the model training expenses. In our case, the AIMACI model necessitates the use of three A100 GPUs for approximately three days to complete the training. Secondly, despite the relatively short training period, the volume of training samples is substantial due to the hourly chemical time step and the fine spatial resolution of our simulation. With 140 by 105 grid cells, 49 vertical layers, and 24 hours in a day, the total number of training samples amounts to 276,595,200 (140x105x49x24x16), reaching the hundred million scale. This large dataset provides a rich and diverse set of samples for training, ensuring that the model does not suffer from a lack of convergence due to insufficient data. However, from the results of our current tests, we acknowledge that increasing the

training dataset, especially by including a broader range of data such as different seasons, could potentially enhance the performance of the AIMACI. Therefore, the 16-day training period represents a balance between computational cost and model performance. In future iterations of AIMACI, we will consider utilizing a larger training dataset to further improve model accuracy and robustness.

Regarding your concern about whether this training period is long enough to capture all the species' residence times or lifetimes, it is important to note that our simulations focus on the concentration changes of aerosols at individual grid points for each time step, without considering transport processes. As such, the residence times/lifetimes of all species were not considered by us.

We have included an explanation of why we chose a relatively short training time window in Section 2.2.2, which reads as follows: "Due to computational cost considerations, the period of the training dataset is not very long; however, the volume of training samples is large due to the hourly chemical time step and the fine spatial resolution of our simulation. With 140 by 105 grid cells, 49 vertical layers, and 24 hours in a day, the total number of training samples amounts to 276,595,200 (140x105x49x24x16), reaching the hundred million scale. This large dataset provides a rich and diverse set of samples for training, ensuring that the AI model does not suffer from a lack of convergence due to insufficient data."

- *12. Results: "The results are promising with an average R² of 0.99 for all 37 evaluated species." Could you have an SI figure plotting the concentration distribution of all species? I wonder if sectional aerosols are easier or harder to emulate if their distributions are normal, flat, or uniform to each other.*

**Response:** Thank you for your valuable suggestion regarding the inclusion of a figure depicting the concentration distribution of all species. We have taken your advice and have added Figure S2, which presents the frequency histograms of the concentration distributions for all output chemical species in the test dataset. This addition has allowed us to further discuss the results. Furthermore, based on another reviewer's suggestion, we have adjusted the output species for the AIMACI scheme and updated all relevant charts in the manuscript. The revised paragraph now reads as follows: "The results are promising, with an average $R^2$ of 0.98 for all 61 evaluated species. This high degree of correlation indicates a strong consistency between the simulations using the AIMACI scheme and the MOSAIC scheme (hereinafter referred to as the numerical scheme). The average NMB for these species is 3.02%, reflecting only a slight deviation from the numerical scheme's outcomes and highlighting the AIMACI scheme's impressive accuracy in simulating ACI. However, as shown in Table 3, some species still exhibit relatively poorer statistical indicators compared to others, such as carbonates. To delve deeper into this observation, we have plotted the frequency histograms of the concentration distributions for all species in the test data (Figure S2). Our analysis revealed that species with skewed concentration distributions, particularly those where more than 99% of the values are close to zero, tend to exhibit poorer statistical indicators. However, this does not signify that the AIMACI scheme has entirely forfeited its predictive capability. As demonstrated in Figure S3, which illustrates the simulated carbonate

[revised manuscript text omitted]

- *13. Results: "3.2 Offline Single-step Simulations with the AIMACI Scheme " -- Should this be Online?*

**Response:** Thank you for your correction. In the revised manuscript, we have corrected this error as "3.2 Online Multi-step Simulations with the AIMACI Scheme".

- *14. Results: Figure 3: Can you provide relative error metrics as well? Seems like AIMACI over/underpredicts SO4 depending on the bin compared to the numerical model. Is there a reason that drives this difference?*

**Response:** Thank you for your suggestion to include relative error metrics in our analysis. As noted in our previous responses (NO.12), calculating relative error metrics for a single time point can be significantly influenced by values close to zero, which may skew the representation of simulation bias. To provide a clearer depiction of the simulation discrepancies, we have opted to present absolute error metrics instead and have accordingly revised Figure 3.

From Figure 3, it can be found that the over/underprediction of $SO_4$ by AIMACI is not bin-dependent but rather value-dependent. For each particle size, the highest concentration regions are predominantly underestimated, while lower values, especially those near zero, are generally overestimated. This pattern may be attributed to two factors: the scarcity of relatively high values in the training data and the use of the Root Mean Square Error (RMSE) as the loss function in our model training. The scarcity of high-value samples may lead to insufficient learning by the model, and the use of RMSE as a loss function may bias the model towards predicting the mean to minimize the loss. The combination of these two factors could potentially contribute to the observed tendencies. Correspondingly, we have incorporated these discussions into the manuscript, as follows:

"From the absolute error figures, it is observed that for each particle size, AIMACI tends to underestimate the higher concentration regions and overestimate lower values, particularly those near zero. This phenomenon may be related to the low proportion of relatively high values in the training dataset and the use of RMSE as the loss function in our model training, which may bias the model towards predicting the mean. Additionally, the results shown are from the last time step of a 10-day continuous simulation, and the simulation errors could be influenced not only by the biases of a single simulation instance but also by potential inaccuracies in the inputs at that time step. Further exploration is necessary to reach a precise conclusion."

[Figure]

**Figure 3:** Sulfate column concentration simulations across different size bins. The first and second column depict the spatial distribution at the 10-day continuous simulation's end (2019-03-30 00:00 UTC), as simulated online by both the numerical scheme and the AIMACI scheme, respectively. The third column is the absolute error between them. The fourth column shows the temporal evolution of the hourly RMSE over the 10-day period. The mean RMSE (unit: mg /m²) for all days and the slope (unit: mg m⁻² h⁻¹) for different simulation stages (2-4day, 5-7day, 8-10day) are given inset.

- *15. Results: "A notable aspect of the AIMACI scheme is its grid-based training and prediction methodology, which contrasts with existing AI large models such as Pangu (Bi et al., 2023) and Fengwu (Chen et al., 2023) that operate on entire fields" This was not stated in the methods, it would help to give more context to the model. Do you not tokenize the entire field?*

**Response:** Thanks for your suggestion. We have added more information about these two methods and revised the text as follows: "In the development of our AIMACI scheme, we faced a bifurcation of choices: In the development of our AIMACI scheme, we faced a bifurcation of choices: whether to input all features of a single grid point and predict for that grid point individually, followed by iterating through all grid points, or to input all features for the entire 3D grid space simultaneously and predict for the entire 3D space at once. Most current AI large models such as Pangu (Bi et al., 2023) and Fengwu (Chen et al., 2023), opt for the latter approach, which inevitably requires the use of convolutional networks. However, in designing the AIMACI scheme, we chose the former method, aligning with the approach taken by numerical models. This method has the advantage of significantly increasing the training sample volume, as each grid point at a given moment constitutes a sample, and it avoids the use of convolutional neural networks, leading to a substantial reduction in computational costs. Moreover, this grid-based AI scheme is versatile, capable of being applied to simulations of regions of any size, without constraints imposed by the size of the training area. However, this approach also presents a challenge in accurately simulating spatial distributions, given the potential for error propagation from neighboring grid points due to physical processes like transport."

- *16. Results: "In Figure 3, the AIMACI scheme has successfully captured and reproduced the intricate spatial patterns of sulfate column concentrations across different particle sizes with R2 values all exceeding 0.88, even after a prolonged 10-day simulation. This achievement underscores the AIMACI scheme's exceptional stability and accuracy. " But the hotspots are over/under-predicted and it seems like RMSE increases after 5 days, and oscillates within each day. Perhaps this error would grow outside of the training time horizon.*

**Response:** Thank you for your comment. We acknowledge the over/under-prediction issues of the AIMACI scheme in multi-step continuous simulations, particularly in the hotspots. While the scheme does capture the overall spatial pattern of sulfate column concentrations effectively, we recognize that there is room for improvement in the accuracy of specific localized predictions. Regarding the RMSE time series, we have recalculated slope of RMSE time series across different simulation stages for four aerosol size bins and added the results in Figure 3. It reveals that the RMSE trends are not uniform across different simulation stages, and even within the same stage, the RMSE trends for different species vary. This indicates that the simulation error for each species is not consistently increasing; there are instances where it decreases. Furthermore, not all species exhibit a simultaneous increase in simulation error; some species show an increase while others show a decrease. This complex error variation may be related to the online continuous simulation approach, as the aerosol concentrations simulated by the AIMACI scheme are subject to other processes such as dry deposition, wet scavenging, and transport. Additionally, in the later sections of the paper, we have extended our analysis to include simulations over a month in different seasons, and these results do not show a significant rapid growth in error, suggesting that the scheme maintains its predictive performance beyond the training time horizon. However, we recognize that our previous description may have been somewhat exaggerated, and we have adjusted the text and add above discussion as follows: "In Figure 3, the AIMACI scheme has effectively captured

the overall spatial distribution of sulfate column concentrations across various particle sizes, with $R^2$ values all exceeding 0.88, even after a 10-day simulation. Although there are instances of underestimation or overestimation, the RMSE time series indicates that the RMSE values remain small throughout the entire simulation period, highlighting the scheme's stability and accuracy. Furthermore, we calculated the slope of RMSE time series across different simulation stages for four aerosol size bins. It reveals that the RMSE trends are not uniform across different simulation stages, and even within the same stage, the RMSE trends for different species vary. This indicates that the simulation error for each species is not consistently increasing; there are instances where it decreases. Furthermore, not all species exhibit a simultaneous increase in simulation error; some species show an increase while others show a decrease. This complex error variation may be related to the online simulation approach, as the aerosol concentrations simulated by the AIMACI scheme are subject to other processes in the numerical model such as dry deposition, wet scavenging."

[Figure]

**Figure 3:** Sulfate column concentration simulations across different size bins. The first and second column depict the spatial distribution at the 10-day continuous simulation's end (2019-03-30 00:00 UTC), as simulated online by both the numerical scheme and the AIMACI scheme, respectively. The third column is the absolute error between them. The fourth column shows the temporal evolution of the hourly RMSE over the 10-day period. The mean RMSE (unit: mg /m$^2$) for all days and the slope (unit: mg m$^{-2}$ h$^{-1}$) for different simulation stages (2-4day, 5-7day, 8-10day) are given inset.

- *17. Results: "suggesting that these discrepancies do not lead to runaway error growth. This sustained performance further substantiates the AIMACI scheme's reliability, positioning it as a robust tool for extended atmospheric and climate simulations. " The language throughout is too strong, championing this model as revolutionary yet it's tested on short time windows. It very well may be, but the results presented in this paper do not warrant this kind of treatment. We do not necessarily expect runaway error growth in 10-day simulations and do not see that is the case in other studies as well. It is over longer-term time scales (e.g., 1 month +) where long-term stability is a concern, whether due to chemical lifetimes cycling over, seasonal weather patterns changing, etc.*

**Response:** Thank you for your suggestion. We have deleted the text to provide a more measured and accurate representation of our findings. The updated text now reads as follows: "In Figure 3, the AIMACI scheme has effectively captured the overall spatial distribution of sulfate column concentrations across various particle sizes, with R$^2$ values all exceeding 0.88, even after a 10-day simulation. Although there are instances of underestimation or overestimation, the RMSE time series indicates that the RMSE values remain small throughout the entire simulation period, highlighting the scheme's stability and accuracy. Furthermore, we calculated the slope of RMSE time series across different simulation stages for four aerosol size bins. It reveals that the RMSE trends are not uniform across different simulation stages, and even within the same stage, the RMSE trends for different species vary. This indicates that the simulation error for each species is not consistently increasing; there are instances where it decreases. Furthermore, not all species exhibit a simultaneous increase in simulation error; some species show an increase while others show a decrease. This complex error variation may be related to the online simulation approach, as the aerosol concentrations simulated by the AIMACI scheme are subject to other processes in the numerical model such as dry deposition, wet scavenging."

- *18. Results: Figure 4. Results here look promising but please provide both absolute error and relative error plots that accompany the figure. Looking at raw concs is not helpful beyond a simple eye test.*

**Response:** Thank you for your suggestion to include both absolute and relative error plots alongside Figure 4. We have revised Figure 4 to include these additional plots. From the new plots, it can be observed that the overall trend in absolute error distribution shows larger errors at the surface and smaller errors at higher altitudes. In contrast, the relative error distribution

follows the opposite pattern, with lower relative errors at the surface and higher relative errors at higher altitudes. This trend is particularly noticeable for nitrate, which, compared to sulfate, exhibits much lower concentrations at higher altitudes—often one or two orders of magnitude smaller, approaching zero. We have incorporated this analysis and discussion into the revised manuscript. The updated text is as follows: "Figure 4 presents a comparison of the zonal average total concentrations (summed across all size bins) of sulfate and nitrate, simulated by both the numerical scheme and the AIMACI scheme, with results averaged over the entire 10-day simulation period. Observations from Figure 4a and 4e indicate that high concentration zones for both sulfate and nitrate are predominantly situated between 25°N and 40°N, coinciding with the latitude range of the Yangtze River Economic Belt. This distribution pattern is likely influenced by the significant anthropogenic emissions in this area. Through turbulent and convective transport processes, sulfate and nitrate from lower altitudes are transported to higher altitudes, with concentrations gradually diminishing with increasing altitude. In Figures 4b and 4f, the AIMACI scheme exhibits a notable alignment with the outcomes from the numerical scheme, as evidenced by the R² values, which are exceptionally high at 0.99 for both sulfate and nitrate. The Root Mean Square Error (RMSE) values are 0.10 µg/kg for sulfate and 0.48 µg/kg for nitrate, suggesting that the discrepancies are minimal, further supporting the AIMACI scheme's accuracy. To provide additional insights, we have also included plots for both the absolute and relative errors. The absolute error distribution shows larger errors near the surface and smaller ones at higher altitudes, while the relative error distribution follows the opposite trend, with lower relative errors near the surface and higher relative errors at greater altitudes. This pattern is particularly prominent for nitrate, which, compared to sulfate, has significantly lower concentrations at higher altitudes—often by one or two orders of magnitude, approaching zero."

[Figure]

**Figure 4:** Zonal mean total concentrations (summed across 4 size bins) of sulfate and nitrate between 109.1°E and 125.9°E, as simulated online by both the numerical scheme and the AIMACI scheme. Results are averages over the entire 10-day simulation period.

- *19. Results: Figure 5. Seems like the AIMACI starts to drift over time (though very small) wonder what happens if simulation time extends longer and longer.*

**Response:** Thank you for your thoughtful review and valuable comments. Indeed, we did not present simulation results beyond the 10-day period in this experiment. However, we later performed separate one-month continuous simulations for January, April, July, and October. Based on the results from the April simulation, we found that the minor discrepancies observed did not lead to a continuous drift in the simulation outcomes. Nevertheless, we fully agree that it is crucial to evaluate the stability of the AIMACI scheme over longer timescales, and we plan to conduct further simulations in the future.

- *20. Results: Figure 7/8: Should also have absolute and relative error plots. Need to see where/why AIMACI is over/under-predicting. It is remarkable that the model has learned the fields well enough to solve for different months. But again: 1) a one month simulation is not long enough to claim that this is long-term stable, especially when we see in the time series there starts to be a drift over time, and 2) there are obvious mismatches of hotspots in the map.*

**Response:** We greatly appreciate your valuable feedback. In response to your suggestions, we have revised Figures 7 and 8 to include absolute and relative error plots and have provided a more detailed discussion. Below is the revised description: "

[revised manuscript text omitted]

**Response:** Thank you for your constructive feedback. As discussed in my previous response (NO.21 ), we have conducted a further investigation into the potential causes of the discrepancies in the simulation results, particularly in July. Our analysis has focused on the impact of meteorological conditions, such as the occurrence of typhoons, which are not well-represented

in our training dataset that consists of partial data from March. The results indicate that specific seasonal weather events, such as typhoons, do indeed significantly impact the performance of the AIMACI scheme. This finding underscores the importance of including a diverse range of meteorological conditions in the training dataset to enhance the AIMACI's robustness and accuracy in simulating atmospheric aerosol chemistry and interactions across different environmental conditions.

- *22. Results: If you could simulate AIMACI for 3 months starting in Spring that would help determine if this model is truly stable. But honestly, a year minimum is what is necessary in the field of atmospheric chemistry and climate, but I do realize that that incurs large computational costs and may be outside of the scope here. But if that is the case, then the discussion of stability needs to be tempered.*

**Response:** Thank you for your valuable suggestion. We acknowledge that conducting longer-term simulations is indeed crucial for assessing the stability of AIMACI. As mentioned earlier, we are actively working to couple both AI-based chemical schemes—photochemistry and aerosol chemistry—within the atmospheric physical-chemical fully coupled global variable-resolution atmospheric numerical model iAMAS, developed by our research group. This integration will enable us to conduct a more comprehensive and nuanced evaluation of AIMACI's performance. We are committed to carrying out these long-term simulation tests and will share the results in our future publications. In response to your concern, we have made adjustments to the manuscript to temper the discussion regarding the stability of the AIMACI scheme.

- *23. Results: "Given that the WRF-Chem, written in Fortran, is not conducive to GPU acceleration, we conducted offline tests of the AIMACI scheme's computational speed on a GPU and compared it with the numerical scheme on a CPU, where the AIMACI scheme was coupled into the WRF-Chem. " Some may argue that this is not an apples-to-apples comparison but I am ok with it. These speedups seem similar to Kelp et al (2020) and Liu et al (2021).*

**Response:** Thank you for your acknowledgment. The speedup of AIMACI observed on a single CPU is indeed similar to the acceleration effects demonstrated in existing AI-based photochemical schemes, such as those in Kelp et al. (2020) and Liu et al. (2021). Currently, we have not made any optimizations to the code, but we plan to further optimize it in the future to improve the speedup.

**Reviewer #2**

*General comments:*

- *This work introduces the Artificial Intelligence Model for Aerosol Chemistry and Interactions (AIMACI), trained on 16 days of WRF-Chem simulation data. When integrated with the online WRF-Chem, AIMACI demonstrates high consistency and accuracy in modeling inorganic aerosols, maintaining stability over one-month scale online simulations. Furthermore, AIMACI exhibits significant computational speedup compared to conventional numerical schemes, highlighting its potential to overcome the computational challenges of traditional methods. The comments listed below are minor clarifications. Once these points are addressed satisfactorily, I believe the paper will be suitable for publication in ACP.*

**Response:** We are grateful for your positive evaluation of our manuscript and appreciate your recognition of the potential of the Artificial Intelligence Model for Aerosol Chemistry and Interactions (AIMACI). Your insightful comments provide valuable guidance for our revisions. We are committed to addressing the clarifications you have highlighted and have made the necessary revisions to enhance both the clarity and quality of the paper. Below, we outline our responses to each of your points to ensure that the manuscript meets the standards for publication in ACP.

*Specific comments:*

- *1. Line 23: "8 aerosol species including water content in aerosols".*

  *-But the main text mainly discusses four species (i.e., $SO_4^{2-}$, $NO_3^-$, aerosol water, and number concentration of aerosol).*

**Response:** We sincerely appreciate your careful review. We acknowledge the confusion that may arise from the mention of "8 aerosol species" in the abstract, while the main text primarily discusses four key species ($SO_4^{2-}$, $NO_3^-$, aerosol water content, and aerosol number concentration). The reason for this is that, due to space constraints, we chose to present the statistical indicators for all species and, considering the correlations between different species, selected four representative species for the spatial and temporal distribution figures. These four species were chosen for their significance in representing the broader trends and interactions within the aerosol population. We have also revised the sentence to more accurately reflect the content of the main text as: "Results demonstrate that AIMACI are not only comparable to those with the conventional scheme in spatial distributions, temporal variations, and evolution of particle size distribution of main aerosol species including water content in aerosols, but also exhibits robust generalization ability, reliably simulating one month under different environmental conditions across four seasons despite being trained on limited data from merely 16 days."

- *2. Line 28-29: "paves the way for its coupling into global climate models for further testing in near future. This*

*advancement promises to enhance the precision and efficiency of atmospheric aerosol simulations in climate modeling*〞.

*-Currently, the AIMAC demonstrates stability only on a one-month scale. Additionally, the training data is derived from a climate model, meaning that AIMAC only reproduces model simulations and does not enhance the precision of the climate model.*

**Response:** Thanks for your comment. We acknowledge that the absence of such long-term validation is a significant limitation. It is for this reason that we are actively working to couple both AI-based chemical schemes—photochemistry and aerosol chemistry—within the atmospheric physical-chemical fully coupled global variable-resolution atmospheric numerical model iAMAS, developed by our research group. This integration will allow us to conduct a more comprehensive and nuanced evaluation of AIMACI's performance over longer timeframes. We fully agree that definitive claims about AIMACI's impact on climate modeling should be reserved until after it has undergone rigorous testing over longer timeframes. To this end, we are committed to carrying out these tests and will share the results in our future publications.

Additionally, as you correctly pointed out, AIMACI in its current version reproduces simulation results and improves computational speed without directly enhancing simulation precision. However, there are many climate models that, due to computational constraints, oversimplify or even turn off the numerical scheme of aerosol chemistry and interactions during long-term high-resolution simulations, introducing significant uncertainties. If AIMACI proves to be stable in year-scale simulations, it can be coupled into these climate models to achieve fast, accurate, and stable simulation of aerosol chemistry and interactions processes, thereby improving simulation precision. Furthermore, in many global variable-resolution numerical models, such as MPAS, atmospheric chemical processes are severely neglected or absent partly due to the extensive modifications required to incorporate them. AIMACI's plug-and-play nature could make the migration of chemical processes to these models much more convenient, thereby enhancing their simulation precision.

We have revised the manuscript to reflect these considerations and to provide a more cautious and evidence-based discussion of AIMACI's potential future impact on climate modeling. The updated abstract now reads: "However, the stability of AIMACI for year-scale global simulations remain to be seen, requiring further testing. AIMACI's generalization capability and its plug-and-play nature suggest potential for future coupling into global climate models, which are expected to enhance the precision and efficiency of aerosol simulations in climate modeling that neglects or simplifies ACI processes."

- *3. Line 38-39 "aerosol chemistry and interactions"*

   *-Please specify which physical and chemical processes are included.*

**Response:** Thank you for your request for clarification on the physical and chemical processes included in "aerosol chemistry and interactions." In response to your comment, we have added the detail description and it now reads: "Aerosol chemistry and interactions (ACI) involve a range of highly nonlinear processes, including chemical reactions and phase equilibrium, gasparticle partitioning, particle size growth, coagulation, and nucleation, which have a significant impact on the concentration of atmospheric aerosols (Zaveri et al., 2008)."

Additionally, while we initially provided this detail in the Methods section (specifically, section 2.2.1 Scheme Construction, lines 132-134), we acknowledge that this information should be more prominently and clearly articulated. Consequently, we have revised it as follows: "In this study, for the first time, we attempt to use an AI algorithm to emulate the sophisticated scheme of ACI (i.e., the MOSAIC scheme), which involves a range of highly nonlinear processes, including chemical reactions and phase equilibrium, gas-particle partitioning, particle size growth, coagulation and nucleation (Zaveri et al., 2008)."

Accordingly, we have also adjusted Figure 1 to make these elements more prominent and clearer.

[Figure]

**Figure 1:** The Artificial Intelligence Model for Aerosol Chemistry and Interactions (AIMACI) in the Weather Research and Forecasting with Chemistry (WRF-Chem). The trained AIMACI is packaged into a static/dynamic library using TorchScript and Libtorch, and can be called by WRF-Chem through an interface to replace the aerosol chemistry and interactions numerical scheme, while the remaining processes maintain the original numerical scheme.

- *4. Line 93-99 "The results demonstrate that, … thereby reducing uncertainties stemming from simplified representations of these processes"*

  *-Results are not recommended to be included in the introduction section.*

**Response:** Thank you for your valuable feedback. We appreciate your suggestion regarding the placement of the results in the introduction. In response to your comment, we have revised the introduction by removing the specific results and instead added a statement to emphasize the validation of AIMACI's accuracy, stability, and computational efficiency. The revised sentence

now reads: "To validate the accuracy, stability, and computational efficiency of the AIMACI scheme, we conducted a series of experiments for both offline simulations (where AIMACI scheme was not coupled to a numerical model) and online simulations (where AIMACI scheme was coupled to a numerical model)."

- *5. Line 116-119: It is recommended to introduce the chemical schemes for sulfate and nitrate. Additionally, how are the emissions of primary aerosols and precursor gases configured in the model?*

**Response:** We appreciate your valuable suggestion. In this study, we have adopted the MOSAIC scheme from Zaveri et al., 2008 without any modifications. As depicted in Table 1 of the referenced paper, our current implementation includes a subset of sulfate and nitrate chemical reactions, such as the interactions between sulfuric acid, nitric acid, and ions like calcium, sodium, and chloride, which lead to the formation of sulfates and nitrates. We have not considered more complex chemical mechanisms at this stage, such as the different heterogeneous formation pathways of sulfates (e.g., oxidation of dissolved S(IV) by $H_2O_2$, $O_3$, $NO_2$ and $O_2$ catalyzed by transition metal ions (TMI) in aerosol water (Ruan et al., 2022)). However, we acknowledge the importance of these reactions and plan to incorporate them in future iterations of our AIMACI model to enhance its comprehensiveness and accuracy. Accordingly, we have added the following description in Section 2.1: "The chemical reactions among various species are detailed in Zaveri et al. (2008). Currently, only a subset of sulfate and nitrate chemical reactions are considered, such as the interactions between sulfuric acid, nitric acid, and ions like calcium, sodium, and chloride, which lead to the formation of sulfates and nitrates. For more complex chemical reactions, such as the different heterogeneous formation pathways of sulfates in aerosol liquid water (e.g., oxidation of dissolved S(IV) by $H_2O_2$, $O_3$, $NO_2$ and $O_2$ catalyzed by transition metal ions (TMI) in aerosol water, (Ruan et al., 2022)), will be incorporated in the future."

Furthermore, for the emissions of primary aerosols and precursor gases, we use the Multi-resolution Emission Inventory for China (MEIC) at a 0.25° x 0.25° horizontal resolution for 2019 (Li et al., 2017a; Li et al., 2017b) within the simulation domain. Similarly, meteorological fields are provided by the National Center for Environmental Prediction (NCEP) final reanalysis (FNL) data, with a 1° x 1° resolution and 6-hour temporal resolution. A detailed description of these datasets has been added in Section 2.2.2 of our manuscript to clarify their configuration in the model, as follows: "For emission and meteorological field, we used the Multi-resolution Emission Inventory for China (MEIC) at 0.25° x 0.25° resolution for 2019 (Li et al., 2017a; Li et al., 2017b), and the NCEP final reanalysis (FNL) data with a 1° x 1° resolution and 6-hour temporal resolution within the simulation domain."

**Table 1.** List of Irreversible Heterogeneous Reactions Included in MOSAIC

| No. | Irreversible Heterogeneous Reactions |
|---|---|
| | *Reactions With $H_2SO_4(g)$* |
| (R1) | $CaCO_3(s) + H_2SO_4(g) \rightarrow CaSO_4(s) + H_2O(g) \uparrow + CO_2(g) \uparrow$ |
| (R2) | $CaCl_2(s,l) + H_2SO_4(g) \rightarrow CaSO_4(s) + 2HCl(g) \uparrow$ |
| (R3) | $Ca(NO_3)_2(s,l) + H_2SO_4(g) \rightarrow CaSO_4(s) + 2HNO_3(g) \uparrow$ |
| (R4) | $2NaCl(s,l) + H_2SO_4(g) \rightarrow Na_2SO_4(s,l) + 2HCl(g) \uparrow$ |
| (R5) | $2NaNO_3(s,l) + H_2SO_4(g) \rightarrow Na_2SO_4(s,l) + 2HNO_3(g) \uparrow$ |
| (R6) | $(CH_3SO_3)_2Ca(s,l) + H_2SO_4(g) \rightarrow CaSO_4(s) + 2CH_3SO_3H(l)$ |
| | *Reactions With $CH_3SO_3H(g)$* |
| (R7) | $CaCO_3(s) + 2CH_3SO_3H(g) \rightarrow (CH_3SO_3)_2Ca(s,l) + H_2O(g) \uparrow + CO_2(g) \uparrow$ |
| (R8) | $CaCl_2(s,l) + 2CH_3SO_3H(g) \rightarrow (CH_3SO_3)_2Ca(s,l) + 2HCl(g) \uparrow$ |
| (R9) | $Ca(NO_3)_2(s,l) + 2CH_3SO_3H(g) \rightarrow (CH_3SO_3)_2Ca(s,l) + 2HNO_3(g) \uparrow$ |
| (R10) | $NaCl(s,l) + CH_3SO_3H(g) \rightarrow CH_3SO_3Na(s,l) + HCl(g) \uparrow$ |
| (R11) | $NaNO_3(s,l) + CH_3SO_3H(g) \rightarrow CH_3SO_3Na(s,l) + HNO_3(g) \uparrow$ |
| | *Reactions With $HNO_3(g)$* |
| (R12) | $CaCO_3(s) + 2HNO_3(g) \rightarrow Ca(NO_3)_2(s) + H_2O(g) \uparrow + CO_2(g) \uparrow$ |
| (R13) | $CaCl_2(s) + 2HNO_3(g) \rightarrow Ca(NO_3)_2(s) + 2HCl(g) \uparrow$ |
| (R14) | $NaCl(s) + HNO_3(g) \rightarrow NaNO_3(s) + HCl(g) \uparrow$ |
| | *Reactions With $HCl(g)$* |
| (R15) | $CaCO_3(s) + 2HCl(g) \rightarrow CaCl_2(s) + H_2O(g) \uparrow + CO_2(g) \uparrow$ |
| | *Reactions With $NH_3(g)$* |
| (R16) | $NH_4HSO_4(s) + NH_3(g) \rightarrow (NH_4)_2SO_4(s)$ |
| (R17) | $(NH_4)_3H(SO_4)_2(s) + NH_3(g) \rightarrow 2(NH_4)_2SO_4(s)$ |
| (R18) | $2NaHSO_4(s) + NH_3(g) \rightarrow Na_2SO_4(s) + NH_4HSO_4(s)$ |

  *-Is the training conducted separately for each grid point? How does the model account for interactions between different grid points?*

**Response:** Thank you for your insightful comment. You are correct. In our study, each training sample indeed corresponds to a single grid point at a given moment, characterized by the aerosol concentration and environmental variables. This is because aerosol chemistry and interactions only include chemical reactions, phase equilibrium, gas-particle partitioning, particle size growth, coagulation, and nucleation, but do not encompass processes like dry deposition, transport, or wet scavenging, as

shown in Figure 1. As a result, the original MOSAIC scheme performs calculations independently for each grid point, without accounting for interactions between different grid points. We have maintained this consistency in our AIMACI scheme.

[Figure]

**Figure 1:** The Artificial Intelligence Model for Aerosol Chemistry and Interactions (AIMACI) in the Weather Research and Forecasting with Chemistry (WRF-Chem). The trained AIMACI is packaged into a static/dynamic library using TorchScript and Libtorch, and can be called by WRF-Chem through an interface to replace the aerosol chemistry and interactions numerical scheme, while the remaining processes maintain the original numerical scheme.

- *7. Line 166-167: "The simulation result was segmented as follows: the initial 16 days from 2019-03-02 00:00 UTC, were designated as the training set"*
  *-Why was such a short training period chosen? Is this the minimum period necessary for training, or does the training cost increase significantly with a longer period?*

**Response:** Thank you for your insightful question regarding the selection of the training period for our simulation. The choice of a 16-day training period was made after careful consideration of several critical factors. Firstly, extending the training period would significantly increase both the numerical simulation costs and the model training expenses. In our case, the AIMACI model necessitates the use of three A100 GPUs for approximately three days to complete the training. Secondly, despite the relatively short training period, the volume of training samples is substantial due to the hourly chemical time step and the fine spatial resolution of our simulation. With 140 by 105 grid cells, 49 vertical layers, and 24 hours in a day, the total number of training samples amounts to 276,595,200 (140x105x49x24x16), reaching the hundred million scale. This large dataset provides a rich and diverse set of samples for training, ensuring that the model does not suffer from a lack of convergence due to insufficient data. However, from the results of our current tests, we acknowledge that increasing the training dataset, especially

by including a broader range of data such as different seasons, could potentially enhance the performance of the AIMACI. Therefore, the 16-day training period represents a balance between computational cost and model performance. In future iterations of AIMACI, we will consider utilizing a larger training dataset to further improve model accuracy and robustness. The corresponding explanation has been added in Section 2.2.2, as shown below: "Due to computational cost considerations, the period of the training dataset is not very long; however, the volume of training samples is large due to the hourly chemical time step and the fine spatial resolution of our simulation. With 140 by 105 grid cells, 49 vertical layers, and 24 hours in a day, the total number of training samples amounts to 276,595,200 (140x105x49x24x16), reaching the hundred million scale. This large dataset provides a rich and diverse set of samples for training, ensuring that the AI model does not suffer from a lack of convergence due to insufficient data."

- *8.Line 172-173: "Each training sample included 65 input features (4 meteorological variables, 5 gas species, and 14 aerosol species with 4 size bins)"*
  *-Precursor gases and atmospheric oxidants (e.g., the hydroxyl radical, ozone) are crucial for the secondary aerosol chemical process. Why are these not included as input features? Additionally, since the training data is hourly, why is the solar zenith angle not considered? Furthermore, the latitude and longitude are also absent.*

**Response:** Thank you for your thoughtful comments. The issue you raised is indeed important. Precursor gases and atmospheric oxidants (such as hydroxyl radicals and ozone) do play a crucial role in secondary aerosol chemistry. However, as I mentioned in my responses to your previous review comment No. 3., our study only focuses on aerosol chemistry and interactions. In the MOSAIC scheme, the gases directly involved in the reactions, such as $H_2SO_4$, $HNO_3$, $HCl$, $NH_3$, and MSA, are indeed included as input features. The production of these gases is closely related to precursor gases and atmospheric oxidants, but the changes and reactions of these components are typically handled in photochemistry, which is calculated separately. The MOSAIC scheme we use does not account for complex heterogeneous processes (such as the oxidation of dissolved S(IV) in aerosol water by $H_2O_2$, $O_3$, $NO_2$, and $O_2$, catalyzed by transition metal ions, as discussed in Ruan et al., 2022). Therefore, precursor gases and atmospheric oxidants are not considered necessary input features in AIMACI.

Regarding the solar zenith angle, if our model were to replace photochemistry, it would indeed be necessary as it is involved in the calculation of photolysis rates, as demonstrated in our other publication (Xia et al., 2024). However, since our focus is on aerosol chemistry and interactions, the solar zenith angle is not a required input feature.

As for the latitude and longitude, the MOSAIC scheme treats each grid point with a cyclic calculation without special processing for specific latitudes and longitudes. Therefore, latitude and longitude are not necessary input features for our model.

Nevertheless, we acknowledge that these variables, as you mentioned, could potentially serve as input features, and their inclusion could improve the model's performance, which would need to be tested.

  *-But coagulation between aerosols can potentially lead to changes in the particle size distribution and also affect the aerosol number concentrations.*

**Response:** Thank you for your astute observation. Previously, we were summing the concentrations of each species across four size bins, which did not capture these subtle changes. Following your comment, we revisited the changes in each size bin of the species (other inorganic mass (OIN), mineral dust, black carbon (BC), organic carbon (OC), calcium ($Ca^{2+}$), and carbonate ($CO_3^{2-}$)) after aerosol chemistry and interactions at each time step. Indeed, we observed changes in particle size distribution, mainly within the 0.039–0.156 μm range (size bin 1). In light of this, we have adjusted the AIMACI output variables and updated all relevant charts in the manuscript. The revised Table 1 and deleted paragraphs are as follows: "~~It should be noted that in the AIMACI scheme's simulation of aerosol processes, the concentrations of other inorganic mass (OIN), mineral dust, black carbon (BC), organic carbon (OC), calcium ($Ca^{2+}$), and carbonate ($CO_3^{2-}$) are not altered. Consequently, these aerosol species are not output variables. However, they play a significant role in affecting the acidity or alkalinity of the atmospheric environment, which in turn influences the formation of aerosols. Therefore, these species are necessary as input variables to ensure that the model accurately reflects the conditions affecting aerosol production.~~"

**Table 2 Input and output variables of Artificial Intelligence Model for Aerosol Chemistry and Interactions (AIMACI)**

| Type | Input variables | Output variables |
|---|---|---|
| Meteorological variables | temperature | - |
| | air density | - |
| | pressure | - |
| | water vapor mixing ratio | - |
| Gas species | $H_2SO_4$ | $H_2SO_4$ |
| | $HNO_3$ | $HNO_3$ |
| | NH3 | $NH_3$ |
| | HCL | HCL |

| | MSA | MSA |
|---|---|---|
| | $SO_4^{2-}$ [Size:1-4] | $SO_4^{2-}$ [Size:1-4] |
| | $NO_3^-$ [Size:1-4] | $NO_3^-$ [Size:1-4] |
| | $NH_4^+$ [Size:1-4] | $NH_4^+$ [Size:1-4] |
| | $Na^+$ [Size:1-4] | $Na^+$ [Size:1-4] |
| | $Cl^-$ [Size:1-4] | $Cl^-$ [Size:1-4] |
| | MSA[Size:1-4] | MSA[Size:1-4] |
| Aerosol species | Water [Size:1-4] | Water [Size:1-4] |
| | Num [Size:1-4] | Num [Size:1-4] |
| | OIN [Size:1-4] | OIN [Size:1-4] |
| | DUST [Size:1-4] | DUST [Size:1-4] |
| | OC [Size:1-4] | OC [Size:1-4] |
| | BC [Size:1-4] | BC [Size:1-4] |
| | $Ca^{2+}$ [Size:1-4] | $Ca^{2+}$ [Size:1-4] |
| | $CO_3^{2-}$ [Size:1-4] | $CO_3^{2-}$ [Size:1-4] |

● *10. Line 177: "they play a significant role in affecting the acidity or alkalinity of the atmospheric environment"*

*-Black carbon and organic carbon do not affect acidity. Why are they included as inputs?*

**Response:** Thank you for your careful review. You are absolutely right, and we apologize for the confusion. The inclusion of black carbon and organic carbon as inputs was not meant to imply that they directly affect the acidity or alkalinity of the atmospheric environment. Instead, their role is in influencing the aerosol surface area, which in turn affects aerosol production. We have removed this text from the manuscript.

● *11. Line 183: "minimizing alterations to the original codebase while offering a lightweight"*

*-It is not clearly specified where the AIMACI scheme is integrated within WRF-Chem. Additionally, it is unclear which specific model processes the AIMACI scheme replaces. What modifications are needed for the interaction and feedback between the AIMACI scheme and subsequent model processes within WRF-Chem?*

**Response:** Thank you for your valuable feedback. We have revised Figure 1 to make all processes more informative and clearer. The updated version explicitly highlights the structure of the AIMACI model, the coupling method, and where AIMACI couples within the WRF-Chem. Additionally, it now details the aerosol chemistry and interactions processes replacing by AIMACI. Regarding your question about the modifications needed for the interaction and feedback between the

AIMACI scheme and subsequent model processes within WRF-Chem, we would like to clarify that no additional modifications are required. This is because AIMACI operates in the same way as the original MOSAIC scheme. It only involves passing the relevant environmental variables and the chemistry species through the interface, and then using the interface to pass the modified chemistry species array after processing.

[Figure]

**Figure 1:** The Artificial Intelligence Model for Aerosol Chemistry and Interactions (AIMACI) in the Weather Research and Forecasting with Chemistry (WRF-Chem). The trained AIMACI is packaged into a static/dynamic library using TorchScript and Libtorch, and can be called by WRF-Chem through an interface to replace the aerosol chemistry and interactions numerical scheme, while the remaining processes maintain the original numerical scheme.

- *12. Line 186: "Furthermore, we conducted three sets of additional experiments"*

  *-It is recommended to distinguish between offline and online AIMACI simulations. Additionally, since there are several numerical experiments, it is suggested to list these experiments clearly in a table.*

**Response:** Thank you for your constructive feedback. Offline simulations involve the use of AIMACI as a standalone model, not coupled within a numerical model, such as WRF-Chem. It operates as a box model on the test set, performing a single-step simulation where a set of input parameters yields a corresponding set of output results. This process is independent of other processes in a numerical model and is used to evaluate the performance of AIMACI in isolation. Online simulations, in contrast, couple AIMACI into a numerical model such as WRF-Chem. In this setup, AIMACI is invoked at each chemical time step during a continuous simulation. The aerosol concentrations generated by AIMACI are influenced by and also influence other atmospheric processes, such as dry deposition. These concentrations are then fed back into AIMACI at next chemical time step, creating a dynamic and iterative simulation of the atmospheric system. Offline and online evaluations are

indeed both necessary and complementary. By conducting both types of evaluations, we can gain a comprehensive understanding of AIMACI's performance. We have also included a table (Table 2) that summarizes the numerical experiments conducted in this study, as you suggested. The new Table 2 and revised paragraph are as follows: "Furthermore, to comprehensively evaluate the performance of the AIMACI scheme, we conducted a series of additional experiments in both offline and online mode, which are detailed in Table 2. These experiments included: (1) Offline simulations were performed on the test dataset without coupling AIMACI scheme into WRF-Chem, treating AIMACI as a standalone box model to assess its single-step performance; (2) Online simulations were conducted for a 10-day continuous period outside the training phase, following the coupling of the AIMACI scheme into WRF-Chem, where it is invoked at each chemical time step and interacts with other model processes to create a dynamic and iterative aerosol simulation; (3) A month-long online continuous simulation was carried out under different environmental conditions across all four seasons to evaluate the AIMACI scheme's generalization ability. To assess the potential impact of the AIMACI scheme on climate research applications, we evaluated statistical indicators for all species and a focused examination of spatial distributions, temporal series, and the evolution of particle size distribution (PSD) for four representative aerosol species. The four selected species include sulfate, nitrate, liquid water content in aerosols, and aerosol number concentration. Sulfate, mainly from fossil fuel emissions, contributes to acid rain, aerosol formation, and aerosol-cloud interactions (Calvert et al., 1985; Fuzzi et al., 2015; Penkett et al., 1979). Nitrate, formed from nitrogen oxides, is a key aerosol component that affects air quality and ecosystems. (Parrish et al., 2012; Saiz-Lopez et al., 2017). Liquid water content in aerosols is important for understanding how particles contribute to cloud formation and precipitation. (Hodas et al., 2014; Liu et al., 2019; Nguyen et al., 2016; Wu et al., 2018). Aerosol number concentration serves as a key metric for assessing aerosol loading and its direct impact on visibility, radiation balance, and climate feedback mechanisms (Spracklen et al., 2010). Collectively, these four species offer a holistic perspective on the multifaceted role of aerosols in atmospheric processes."

**Table 2:** Numerical experiments conducted in this study

| Number | ACI Scheme | Period (Hourly) | Type |
|---|---|---|---|
| EXP 0 | MOSAIC | Mar 1st ~ Mar 19th (Train: Mar 2nd ~ Mar 17th, Validation: Mar 18th, Test: Mar 19th) | Online Continuous Simulation |
| EXP 1 | AIMACI | Mar 19th | Offline Single-step Simulation |
| EXP 2&3 | MOSAIC & AIMACI | Mar 20th ~ Mar 30th | Online Continuous Simulation |
| EXP 4&5 | MOSAIC & AIMACI | Jan, Apr, Jul, Oct (1 month) | Online Continuous Simulation |

- *13. Line 232-237: "Atmospheric aerosols significantly impact the climate system … climate models for precise climate simulations"*

    *-It does not add much value. It is recommended to provide a more detailed discussion on the results, such as comparing the spatial distribution correlation between the AIMACI and numerical schemes.*

**Response:** Thank you for your valuable suggestion. We have also adjusted this section to focus more on the discussion of the results. The revised text is as follows: "The results are promising, with an average $R^2$ of 0.98 for all 61 evaluated species. This high degree of correlation indicates a strong consistency between the simulations using the AIMACI scheme and the MOSAIC scheme (hereinafter referred to as the numerical scheme). The average NMB for these species is 3.02%, reflecting only a slight deviation from the numerical scheme's outcomes and highlighting the AIMACI scheme's impressive accuracy in simulating ACI. However, as shown in Table 3, some species still exhibit relatively poorer statistical indicators compared to others, such as carbonates. To delve deeper into this observation, we have plotted the frequency histograms of the concentration distributions for all species in the test data (Figure S2). Our analysis revealed that species with skewed concentration distributions, particularly those where more than 99% of the values are close to zero, tend to exhibit poorer statistical indicators. However, this does not signify that the AIMACI scheme has entirely forfeited its predictive capability. As demonstrated in Figure S3, which illustrates the simulated carbonate concentrations in the 0.625–2.5 µm particle size range ($CO_3\_a03$), the AIMACI scheme continues to perform well in predicting concentration changes in high-value regions. The poorer statistical indicators are primarily attributed to the challenge of accurately forecasting the very low values that are close to zero."

**Table 3:** Statistical metrics on the test dataset of Artificial Intelligence Model for Aerosol Chemistry and Interactions (AIMACI) (The RMSE of different species has different unit: aerosol (µg/kg), num (kg-1), gas (ppmv)).

| Number | Variable | $R^2$ | RMSE | NMB (%) | Number | Variable | $R^2$ | RMSE | NMB (%) |
|---|---|---|---|---|---|---|---|---|---|
| 1 | $H_2SO_4$ | 0.97 | 2.99E-07 | -2.10 | 32 | $Ca\_a02$ | 0.95 | 2.32E-05 | -2.87 |
| 2 | $HNO_3$ | 1.00 | 3.61E-05 | 0.19 | 33 | $CO_3\_a02$ | 0.61 | 2.90E-05 | 28.40 |
| 3 | $NH_3$ | 1.00 | 4.84E-05 | 3.49 | 34 | $SO_4\_a03$ | 0.94 | 2.90E-02 | 0.80 |
| 4 | HCL | 1.00 | 9.68E-06 | -0.41 | 35 | $NO_3\_a03$ | 1.00 | 2.68E-02 | 0.89 |
| 5 | MSA | 0.82 | 1.03E-09 | 0.12 | 36 | $NH_4\_a03$ | 1.00 | 5.37E-03 | 0.48 |
| 6 | $SO_4\_a01$ | 1.00 | 1.14E-02 | 0.16 | 37 | $Na\_a03$ | 1.00 | 2.01E-03 | 2.44 |
| 7 | $NO_3\_a01$ | 1.00 | 4.90E-02 | -0.43 | 38 | $Cl\_a03$ | 1.00 | 4.98E-03 | 1.29 |
| 8 | $NH_4\_a01$ | 1.00 | 1.44E-02 | -0.45 | 39 | $MSA\_a03$ | 1.00 | 2.94E-06 | 2.85 |
| 9 | $Na\_a01$ | 1.00 | 6.10E-06 | 0.32 | 40 | $Water\_a03$ | 0.99 | 4.11E-01 | 2.03 |

| | | | | | | | | | |
|---|---|---|---|---|---|---|---|---|---|
| 10 | Cl_a01 | 0.99 | 2.52E-03 | -1.42 | 41 | Num_a03 | 1.00 | 1.05E+05 | 0.45 |
| 11 | MSA_a01 | 1.00 | 1.78E-05 | 5.70 | 42 | OIN_a03 | 1.00 | 1.86E-02 | 6.34 |
| 12 | Water_a01 | 1.00 | 5.60E-01 | 0.46 | 43 | DUST_a03 | 1.00 | 1.30E-01 | 2.30 |
| 13 | Num_a01 | 1.00 | 4.45E+07 | -0.10 | 44 | OC_a03 | 1.00 | 1.37E-02 | -0.45 |
| 14 | OIN_a01 | 1.00 | 7.98E-03 | 4.36 | 45 | BC_a03 | 1.00 | 2.61E-03 | -0.53 |
| 15 | DUST_a01 | 0.95 | 2.47E-04 | 3.80 | 46 | Ca_a03 | 1.00 | 5.28E-04 | 2.62 |
| 16 | OC_a01 | 1.00 | 4.97E-03 | 0.80 | 47 | $CO_3$_a03 | 0.90 | 1.42E-03 | 88.17 |
| 17 | BC_a01 | 1.00 | 1.49E-03 | -0.64 | 48 | $SO_4$_a04 | 1.00 | 7.25E-04 | 0.27 |
| 18 | Ca_a01 | 0.95 | 9.93E-07 | -0.49 | 49 | $NO_3$_a04 | 0.99 | 5.11E-02 | 0.75 |
| 19 | $CO_3$_a01 | 0.79 | 5.02E-12 | 0.09 | 50 | $NH_4$_a04 | 0.98 | 5.62E-03 | -3.21 |
| 20 | $SO_4$_a02 | 1.00 | 4.16E-02 | 0.38 | 51 | Na_a04 | 1.00 | 9.19E-03 | 4.13 |
| 21 | $NO_3$_a02 | 1.00 | 7.04E-02 | 0.69 | 52 | Cl_a04 | 1.00 | 1.13E-02 | 3.28 |
| 22 | $NH_4$_a02 | 1.00 | 2.31E-02 | 0.51 | 53 | MSA_a04 | 0.79 | 2.00E-05 | 7.93 |
| 23 | Na_a02 | 1.00 | 1.65E-04 | 0.10 | 54 | Water_a04 | 0.99 | 1.35E+00 | -3.32 |
| 24 | Cl_a02 | 1.00 | 4.63E-03 | -1.25 | 55 | Num_a04 | 1.00 | 5.89E+03 | 2.86 |
| 25 | MSA_a02 | 1.00 | 1.51E-05 | 3.05 | 56 | OIN_a04 | 1.00 | 2.27E-02 | 3.93 |
| 26 | Water_a02 | 0.99 | 1.32E+00 | 1.58 | 57 | DUST_a04 | 1.00 | 8.85E-01 | -4.84 |
| 27 | Num_a02 | 1.00 | 9.20E+06 | 0.61 | 58 | OC_a04 | 0.96 | 4.72E-04 | -1.38 |
| 28 | OIN_a02 | 1.00 | 4.49E-02 | -1.80 | 59 | BC_a04 | 0.95 | 1.09E-04 | 5.14 |
| 29 | DUST_a02 | 1.00 | 5.25E-03 | -0.39 | 60 | Ca_a04 | 1.00 | 3.24E-03 | 1.51 |
| 30 | OC_a02 | 1.00 | 4.23E-02 | 2.67 | 61 | $CO_3$_a04 | 0.95 | 1.39E-02 | 15.00 |
| 31 | BC_a02 | 1.00 | 1.12E-02 | -2.54 | Average $R^2$: 0.98 | | | Average NMB: 3.02% | | |

[Figure]

**Figure S2:** Frequency distribution of concentrations for all output chemical species in the test dataset.

[Figure]

**Figure S3:** The spatial distribution of carbonate surface concentrations in the 0.625–2.5 µm particle size range (CO$_3$_a03) simulated by both MOSAIC and AIMACI at 2019-03-19-16:00:00(UTC).

- *14. Line 244-252: "Sulfate, derived primarily from … role in atmospheric processes"*

  *-This is not a description of the results but rather a lot of background information. It is recommended to not include it in this section.*

**Response:** We extend our gratitude for your insightful suggestion. Following your guidance, we have relocated the text to

Section 2.2.2, "Training and Testing Procedure," where it is more fittingly situated within the framework of our methodological approach.

- **15. Line 266: "3.2 Offline Single-step Simulations with the AIMACI Scheme"**

  **-Online AIMACI simulation?**

**Response:** Thank you for your correction. In the revised manuscript, we have corrected this error as "3.2 Online Multi-step Simulations with the AIMACI Scheme".

- **16. Line 286: "Figure 3" & Line 307 "Figure 4"**

  **-It is recommended to supplement the spatial distribution of relative errors.**

**Response:** Thank you for your suggestion to include relative error metrics in our analysis. As noted in our previous responses (NO.13), calculating relative error metrics for a single time point can be significantly influenced by values close to zero, which may skew the representation of simulation bias. To provide a clearer depiction of the simulation discrepancies, we have opted to present absolute error metrics in revised Figure 3 (the spatial distribution at a single time point) and to present both absolute error and relative errors metrics in revised Figure 4 (the monthly average simulation results). Accordingly, we have also adjusted the original description as follows:

[revised manuscript text omitted]

    *-What is the grid-based methodology? How does it differ from large models?*

**Response:** Thanks for your suggestion. We have added more information about these two methods and revised the text as follows: "In the development of our AIMACI scheme, we faced a bifurcation of choices: whether to input all features of a single grid point and predict for that grid point individually, followed by iterating through all grid points, or to input all features for the entire 3D grid space simultaneously and predict for the entire 3D space at once. Most current AI large models such as Pangu (Bi et al., 2023) and Fengwu (Chen et al., 2023), opt for the latter approach, which inevitably requires the use of convolutional networks. However, in designing the AIMACI scheme, we chose the former method, aligning with the approach taken by numerical models. This method has the advantage of significantly increasing the training sample volume, as each grid point at a given moment constitutes a sample, and it avoids the use of convolutional neural networks, leading to a substantial reduction in computational costs. Moreover, this grid-based AI scheme is versatile, capable of being applied to simulations of regions of any size, without constraints imposed by the size of the training area. However, this approach also presents a challenge in accurately simulating spatial distributions, given the potential for error propagation from neighboring grid points due to physical processes like transport."

- *18. Line 305-306 "This sustained performance further substantiates the AIMACI scheme's reliability, positioning it as a robust tool for extended atmospheric and climate simulations."*

*-Please be careful in your discussions. The results in Figure 3 are based on only 10 days of testing. They do not prove reliability for climate simulations.*

**Response:** Thank you for your constructive feedback. We have deleted the text to provide a more measured and accurate representation of our findings. The updated text now reads as follows: "In Figure 3, the AIMACI scheme has effectively captured the overall spatial distribution of sulfate column concentrations across various particle sizes, with $R^2$ values all exceeding 0.88, even after a 10-day simulation. Although there are instances of underestimation or overestimation, the RMSE time series indicates that the RMSE values remain small throughout the entire simulation period, highlighting the scheme's stability and accuracy. Furthermore, we calculated the slope of RMSE time series across different simulation stages for four aerosol size bins. It reveals that the RMSE trends are not uniform across different simulation stages, and even within the same stage, the RMSE trends for different species vary. This indicates that the simulation error for each species is not consistently increasing; there are instances where it decreases. Furthermore, not all species exhibit a simultaneous increase in simulation error; some species show an increase while others show a decrease. This complex error variation may be related to the online simulation approach, as the aerosol concentrations simulated by the AIMACI scheme are subject to other processes in the numerical model such as dry deposition, wet scavenging."

● *19. Line 321 Figure 5: It is recommended to add hourly ticks in Figure 5. Additionally, why is there no apparent diurnal cycle pattern?*

**Response:** Thank you for your valuable suggestion regarding Figure 5. We have taken your advice and revised Figure 5 to include grey vertical lines, which mark the time intervals from 0 to 24 hours in Beijing Time for each corresponding day. From Figure 5, we observe that the diurnal cycle pattern is not immediately apparent on the first day of the simulation, which we attribute to the model being in a spin-up phase. However, as the simulation progresses, the diurnal cycle patterns become increasingly evident. To further illustrate this, we have selected a specific day and plotted the time series of surface total concentrations of sulfate for that day, as shown below. The diurnal cycle pattern is clearly visible, demonstrating the AIMACI's ability to capture the daily variations in concentration.

[Figure]

**Figure R1:** Diurnal variation of surface total concentrations of sulfate, as simulated by the numerical scheme and the AIMACI

scheme.

[Figure]

**Figure 5:** Time series of surface total concentrations (summed across 4 size bins) of four key aerosol species (sulfate, nitrate, liquid water content of aerosol, and number concentration of aerosol), as simulated online by both the numerical scheme and the AIMACI scheme. Results represent the calculated averages for the Yangtze River Delta region (119.1°E~121.9°E, 30.1°N~31.9°N). The grey vertical lines mark the time intervals from 0 to 24 hours in Beijing Time for each corresponding day.

- **20. Line 328 "Despite these pronounced fluctuations"**

  *-It seems that fluctuations increase after the fifth day (e.g., sulfate). What could be the possible reasons for this?*

**Response:** Thank you for your insightful comment. Essentially, the errors in a single timestep simulation at a given grid point are influenced by both the inputs to AIMACI at that timestep and the performance of the AIMACI model itself. One possibility is that the inputs for a particular timestep could be completely accurate, but the AIMACI model has inherent limitations, such as a limited ability to learn special combinations of input features. This could lead to minor discrepancies in the simulation

output even when the inputs are correct. On the other hand, it is also possible that the inputs themselves contain biases, which would then propagate into the model's simulation results. The latter scenario is more common in online continuous simulations. There are two primary reasons for input biases at each timestep. First, errors at the grid point could accumulate over time, which affects subsequent inputs. Second, biases in neighboring grid points may propagate through the numerical model's other processes, such as transport, and influence the inputs for the target grid point. In practice, when running a physics-AI hybrid model for online continuous simulations, these error sources are often interrelated rather than independent, making the analysis of error sources quite complex and necessitating further investigation. We will continue to explore and refine these aspects to better understand and mitigate such fluctuations.

- *21. Line 372 "Figure 7"*

  *-Is it an online AIMACI simulation? Please clarify.*

**Response:** Yes, you are correct. This is an online simulation, where the AIMACI scheme is coupled with the numerical model. Using this physics-AI hybrid model, we simulate one month under different seasonal environmental conditions. The corresponding figure title is modified to: Figure 7: Monthly average surface total concentrations (summed across 4 size bins) of nitrate for different environmental conditions across seasons, as simulated online by both the numerical scheme and the AIMACI scheme.

- *22. Line 381-383 "The AIMACI scheme, despite being trained on data from only 16 days in March, demonstrates a remarkable ability to reproduce these distribution characteristics across different environmental conditions"*

  *-Why can AIMACI scheme trained from March generalize to other seasons? Does your training data sufficiently cover these scenarios, or does your model have some extrapolation capability?*

**Response:** Thank you for your comment. We all know that in the numerical model, no matter which season it is, the same numerical scheme is used to calculate aerosol and chemistry processes, and there is no special treatment for a certain season. Therefore, in theory, if the AIMACI scheme has sufficiently learned the full range of aerosol and chemistry processes from the training data in March, it should be able to generalize to other months as well. However, as you rightly point out, there are limitations to this learning capability. Using more diverse training data and employing more complex model architectures could potentially improve these limits, but achieving perfect generalization remains challenging. Our test results show that AIMACI performs better in January, April, and October, while its performance is relatively weaker in July. This indicates that while AIMACI can generalize to some extent, seasonal variations, especially extreme conditions like those in July, pose a greater challenge for the model. To surmount these limitations and bolster the precision of the AIMACI scheme, future iterations should consider integrating a more comprehensive and diverse training dataset that accounts for a broader spectrum of environmental conditions, notably incorporating additional seasonal meteorological phenomena such as typhoons. The

augmentation of the training dataset would facilitate the fine-tuning of the AIMACI scheme, thereby enhancing its reliability in delivering robust simulation outcomes.

- *23. Line 403-404 "This phenomenon is closely related to the interactions between aerosols and other processes within the physics-AI hybrid model, which differ significantly from offline simulation scenarios"*
  *-I would like the authors to provide a more detailed explanation, such as comparing the differences between offline and online AIMACI schemes.*

**Response:** We greatly appreciate your valuable feedback. In response to your suggestions, we have revised Figures 7 and 8 to and have provided a more detailed discussion. Below is the revised description: "

[revised manuscript text omitted]

---

## Author Response (AR2)

**Editor**

*General comments:*

● *Public justification (visible to the public if the article is accepted and published):*
*One reviewer further raised some critical comments for the revised manuscript. In particular, the reviewer was not satisfied with the presentation of the manuscript. Further improvement in the presentation of the manuscript is needed.*

● *Notification to the authors:*
*Coloured or marked text in \*.pdf supplement file is not allowed. Please provide a clean version of the \*.pdf supplement file (with black text) with the next revision. I would recommend you combine the coloured version of the supplement and the coloured version of the manuscript into one Author's tracked changes version of the files. This way you will be able to save data about changes both in the manuscript and in the supplement.*

Dear Editors:

We sincerely appreciate your editorial effort and the constructive comments provided by the reviewer. We have carefully addressed all the concerns raised, particularly those regarding the presentation of the manuscript, to further improve its clarity and readability.

Additionally, as requested, we have provided a clean version of the supplementary file (with black text only) and merged the tracked changes from both the manuscript and the supplementary materials into a single Author's tracked changes file for clearer documentation of revisions.

Thank you for your time and valuable guidance. We hope the revised manuscript now meets the journal's standards.

Yours sincerely,
Zihan Xia and co-authors

**Reviewer #1**

*Specific Comments:*

- *1. The term 'plug-and-play' is not scientific, rephrase or remove it as it sounds superfluous.*

**Response:** Thank you for your valuable feedback on the use of the term "plug-and-play" in our manuscript. We agree that the phrasing may come across as informal and have revised it to better align with scientific terminology. In the revised manuscript, we have replaced "plug-and-play nature" with "modular design" to more accurately convey the model's adaptability and ease of implementation within broader climate frameworks. The updated text in the Abstract (Lines 27-29) now reads:

"AIMACI's generalization capability and its modular design suggest potential for future coupling into global climate models, which are expected to enhance the precision and efficiency of aerosol simulations in climate modeling that neglects or simplifies ACI processes."

- *2. "After that, the MOSAIC scheme is employed to address aerosol chemistry and interactions. The detailed processes included in aerosol chemistry and interactions are chemical reactions and phase equilibrium, gas-particle partitioning, particle size growth, coagulation, and nucleation, as explained in my response to your previous review comment No. 3."*

  *-->This is still vague. your review comments are directed at me but should be directed towards the text. You write that "Aerosol chemistry and interactions (ACI) involve a range of highly nonlinear processes, including chemical reactions and phase equilibrium, gas-particle partitioning, particle size growth, coagulation, and nucleation ...", so aerosol chemistry is the entire WRF-Chem chemistry model? Are you just emulating and replacing MOSAIC? You need to explicitly state what components you are replacing instead of referring to general physical/chemical processes like coagulation.*

**Response:** Thank you for highlighting the need for clarity regarding the scope of AIMACI within the WRF-Chem. We apologize for any ambiguity in our previous manuscript version.

As shown in Figure 1, in the WRF-Chem chemistry model, the Aerosol Chemistry and Interactions (ACI) is one of several components (e.g., emissions, photolysis, photochemistry, cloud chemistry, etc.). The full ACI consists of multiple subprocesses, including chemical reactions, phase equilibrium, gas-particle partitioning, particle size growth, coagulation, and nucleation. The MOSAIC scheme within WRF-Chem can be used to simulate the full ACI. In our study, the chosen MOSAIC scheme features four discrete size bins and primarily focuses on processes related to inorganic aerosols. It also considers the impact of marine biogenic sources of dimethyl sulfide on atmospheric aerosols and some aqueous reactions. However, secondary organic aerosols and complex heterogeneous chemical processes (e.g., oxidation of dissolved S(IV) by $H_2O_2$, $O_3$, $NO_2$ and $O_2$ catalyzed

by transition metal ions (TMI) in aerosol water, (Ruan et al., 2022)) are not included in the chosen MOSAIC scheme in this study. These aspects will be considered in future development. The AIMACI scheme developed by us is used to achieve end-to-end simulation of the full ACI within a 3D atmospheric numerical model, replacing the chosen MOSAIC numerical scheme to improve computational efficiency.

To emphasize this point, we have revised the relevant text and added a pseudo-code diagram, as follows:

➢ Section 1 (Lines 37-45)

[revised manuscript text omitted]

(a) Pseudo-code for original chemistry model  (b) Pseudo-code for AI-mixed chemistry model

```
subroutine chem_driver
    defining interfaces
    defining variablers
    initializing data
    for each tile
        call emissions_driver
        call optical_driver
        call photolysis_driver
        call dry_dep_driver
        call grelldrvct
        call mechanism_driver
        call cloudchem_driver
        call aerosols_driver
            call mapaer_tofrom_host
            call mosaic_dynamic_solver
            call move_sections
            call mosaic_newnuc_1clm
            call mosaic_coag_1clm
            call mapaer_tofrom_host
        call wetscav_driver
        call budget_calc
    end for
end subroutine chem_driver
```

```
subroutine chem_driver
    defining interfaces
    defining variablers
    initializing data
    for each tile
        call emissions_driver
        call optical_driver
        call photolysis_driver
        call dry_dep_driver
        call grelldrvct
        call mechanism_driver
        call cloudchem_driver
        call AIMACI
        call wetscav_driver
        call budget_calc
    end for
end subroutine chem_driver
```

**Figure S2:** Schematic diagram of the specific components replaced by the AIMACI scheme in WRF-Chem. (a) The pseudo code for original chemistry model in WRF-Chem. The part surrounded by the blue dotted box is the key subroutine inside aerosols_driver. (b) The pseudo code for AI-mixed chemistry model in WRF-Chem. The AIMACI scheme enables end-to-end simulation of full aerosol chemistry and interactions processes, including chemical reactions, phase equilibrium, gas-particle partitioning, particle size growth, coagulation, and nucleation. Note that in our study, the chosen MOSAIC scheme features four discrete size bins and primarily focuses on processes related to inorganic aerosols. It also considers the impact of marine biogenic sources of dimethyl sulfide on atmospheric aerosols and some aqueous reactions. However, secondary organic aerosols and complex heterogeneous chemical processes are not included in the chosen MOSAIC scheme. These aspects will be considered for future development.

- *3. "As the first step, this study focuses on inorganic aerosols, because the chemistry of organic aerosols (i.e., secondary organic aerosols) still has large uncertainties and lacks a convincing numerical scheme for AI scheme to emulate"*

  *--> There are many SOA numerical schemes that exist. I do not know how you can say there is a lack of a convincing scheme.*

**Response:** Thank you for raising this important point and sorry for the inappropriate statement. We acknowledge that there are existing numerical schemes for secondary organic aerosols (SOA) that have been widely implemented in atmospheric models. Our original phrasing regarding the "lack of a convincing scheme" was imprecise and has been revised to better reflect the motivation for prioritizing inorganic aerosols in this initial study. In the revised manuscript (Section 1, Lines 97-99), the text now states:

"As a first step, this study primarily focuses on inorganic aerosols because they constitute significant amounts of secondary aerosols globally and serve as the important driver of aerosol radiative forcing and cloud condensation nuclei activity in current climate models. Given that the production of secondary organic aerosols (SOA) involves significantly more complex chemical pathways, encompassing a wider array of precursor species and heterogeneous reaction mechanisms, the current AIMACI does not include them. Their incorporation will be considered in the future development of AIMACI."

- *4. " its utilization in ACI simulation represents a new horizon. The algorithm's ability to globally attend to input variables and conduct parallel computations across multiple heads is pivotal in tackling the challenges posed by the curse of dimensionality, capturing complex interdependencies, and significantly enhancing computational efficiency."*

  *Again, tone down this language. This is a scientific paper, not an infomercial. The results are promising but it's more of a demo than a useable model, only trained on a tiny amount of useable data.*

**Response:** Thank you for your valuable feedback. We have removed this description and revised the relevant text to reflect a more measured and scientific tone. The revised text now reads as follows (Section 2.2.1, Lines 139-149):

"In this study, for the first time, we attempt to use an AI algorithm to achieve end-to-end simulation of the full ACI within a 3D atmospheric numerical model, replacing the chosen MOSAIC numerical scheme to improve computational efficiency. Given the complexity of multiple subprocesses involved in full ACI, there is a clear need for AI algorithms with superior representational capacity for nonlinear systems. The Multi-Head Self-Attention (MHSA) mechanism, a pivotal component of state-of-the-art transformer architectures, has demonstrated exceptional performance across diverse domains such as Natural Language Processing (Vaswani et al., 2017), Computer Vision (Liu et al., 2021), and Weather Forecasting (Bi et al., 2023). Additionally, in the development of an Artificial Intelligence PhotoChemistry scheme, Xia et al. (2024) have highlighted that the MHSA algorithm excels in capturing the intricate chemical relationships among different species through calculating attention weights. It offers not only high accuracy and computational efficiency but is also less susceptible to the increase in the number of chemical species. Building upon these advancements, this study leverages the MHSA algorithm to develop the Artificial Intelligence Model for Aerosol Chemistry and Interactions (AIMACI)."

- *5. "After training, the AIMACI scheme was flexibly coupled into WRF-Chem"*

  *-->Not sure what flexibly means here, vague*

**Response:** Thank you for your comments. What we intended to convey is that the modular design and lightweight interface of the AIMACI scheme make it easy to couple with WRF-Chem. Considering that we have introduced the advantages of our coupling method over other third-party libraries in detail, we delete "flexibly" here and revised the relevant text, as follows (Section 2.2.1, Lines 163-173):

"The trained AIMACI scheme was packaged into a static (or dynamic) library and then coupled into WRF-Chem, utilizing TorchScript and Libtorch tools officially provided by PyTorch. Compared to coupling approaches relying on third-party libraries such as CFFI (C Foreign Function Interface for Python), our method demonstrates two distinct technical advantages: (1) Enhanced Computational Efficiency: TorchScript and LibTorch, as core components of the PyTorch ecosystem, are specifically optimized for AI model deployment in C++ environments. This optimization reduces computational overhead compared to generic third-party libraries. (2) Streamlined Implementation: PyTorch's official documentation provides standardized workflows for serializing trained AI models into static or dynamic libraries via TorchScript and LibTorch. In contrast, third-party libraries lack native support for AI model deployment, necessitating manual reimplementation of low-level interfaces. Therefore, our coupling approach minimizes alterations to the original codebase and offers a lightweight, adaptable, and modular solution. It is capable of encapsulating a wide range of complex AI algorithms and coupling them with atmospheric and climate models."

- *6. "Secondly, in terms of compatibility, LibTorch supports cross-platform deployment, offering more flexibility than third-party libraries. "*

  *--> This does not sound scientific, sounds like a commercial. Cross-platform is vague in the context of WRF-Chem and scientific computing*

**Response:** Thank you for pointing out this. We have deleted this description and adjusted the relevant text, as follows (Section 2.2.1, Lines 165-173):

"Compared to coupling approaches relying on third-party libraries such as CFFI (C Foreign Function Interface for Python), our method demonstrates two distinct technical advantages: (1) Enhanced Computational Efficiency: TorchScript and LibTorch, as core components of the PyTorch ecosystem, are specifically optimized for AI model deployment in C++ environments. This optimization reduces computational overhead compared to generic third-party libraries. (2) Streamlined Implementation: PyTorch's official documentation provides standardized workflows for serializing trained AI models into static or dynamic libraries via TorchScript and LibTorch. In contrast, third-party libraries lack native support for AI model deployment, necessitating manual reimplementation of low-level interfaces. Therefore, our coupling approach minimizes alterations to the original codebase and offers a lightweight, adaptable, and modular solution. It is capable of encapsulating a

wide range of complex AI algorithms and coupling them with atmospheric and climate models."

● *7. "The average NMB for these species is 3.02%, reflecting only a slight deviation from the numerical scheme's outcomes and highlighting the AIMACI scheme's impressive accuracy in simulating ACI. However, as shown in Table 3, some species still exhibit relatively poorer statistical indicators compared to others, such as carbonates. "*

*--> The manuscript still has way too many value judgments throughout using the terms 'impressive' and 'promising'. You say something is impressive and then the next line you say some parts do poorly. Please take out these instances, they are not scientific and detract from the quality of the paper. Some elements are indeed impressive, but at the end of the day, you are using a relatively simple transformer model trained on 2 weeks of data which does not feel particularly 'impressive' from the ML-weather and ML-NLP communities. Please temper this kind of unscientific language*

**Response:** We sincerely appreciate your rigorous critique regarding the use of subjective language in our manuscript. We fully agree that value judgments like "impressive" and "promising" detract from scientific objectivity. We have systematically revised the text to eliminate such terminology and reframe the discussion around quantitative metrics, as follows (Section 3.1, Lines 265-269):

"Quantitative evaluation across 61 output targets demonstrates close alignment between the simulations using the AIMACI scheme and the MOSAIC scheme (hereinafter referred to as the numerical scheme), as evidenced by an average $R^2$ of 0.98 and an average NMB of 3.02%. While 55 output targets, including major inorganic aerosols such as sulfate, nitrate, and ammonium, achieve $R^2$ values ≥0.95, there are still a few species exhibit relatively poorer statistical metrics, particularly carbonates (Table 3)."

● *8. "From the absolute error figures, it is observed that for each particle size, AIMACI tends to underestimate the higher concentration regions and overestimate lower values, particularly those near zero."*

*-->Same findings in Kelp et al. (2022) which uses much simpler ML methods. Why do you mention the loss function (RMSE)? What loss can you change it to for better results?*

**Response:** Thanks for your insightful comment. The systematic bias pattern observed in our results indeed aligns with the findings of Kelp et al. (2022), who reported similar behavior in MSE-optimized emulators. The tendency of the AI model to underestimate high values and overestimate low values can be related to the mathematical formulation of the RMSE (or MSE) loss function. Specifically, the quadratic term in RMSE (or MSE) penalizes large errors more heavily, which biases the model towards predicting the mean of the target distribution to minimize the overall loss (Gneiting, T., 2011).

[revised manuscript text omitted]

  *--> Vague reason, what do inaccuracies in the inputs at that time step even mean? The ground truth is the WRF-Chem simulation*

**Response:** Thank you for your comment. In Figure 3, we compare the results of the last time step from a 10-day continuous simulation using the AIMACI scheme and the MOSAIC scheme. While both AIMACI and MOSAIC schemes operate within the same WRF-Chem framework with identical meteorological/emission inputs initially, there are slight deviations in the concentrations of different chemical species simulated by the two schemes during the continuous simulation. Therefore, there are certain differences in the inputs of the two schemes in the last time step of the simulation. This is what we originally intended to convey by "potential inaccuracies in the inputs at that time step." However, we agree that this explanation is vague and may not be necessary, as the ground truth for comparison is indeed the WRF-Chem simulation itself. Therefore, we have removed this statement from the manuscript to improve clarity and focus on the primary findings.

- *10. "This complex error variation may be related to the online simulation approach, as the aerosol concentrations simulated by the AIMACI scheme are subject to other processes in the numerical model such as dry deposition, wet scavenging."*

  *--> You cannot say this without showing this. It comes off as deflecting the deficiencies of the ML model.*

**Response:** Thank you for pointing this out. We have reorganized the language in this section to acknowledge the limitations of the AIMACI scheme and analyze the possible reasons for the complex error variation, as shown below (Section 3.2.1, Lines 338-354):

"Additionally, RMSE temporal evolution reveals that during the entire simulation period, it does not exhibit rapid growth but maintains oscillations within a constrained range. The analysis of the RMSE trend slopes across different simulation stages for four size bins demonstrates non-uniform error progression patterns (e.g., in Figure 3l, the slope for the 2–4 day period is 0.000959 mg m$^{-2}$ h$^{-1}$, while the slope for the 8–10 day period is -0.000127 mg m$^{-2}$ h$^{-1}$), where even within identical simulation phases, distinct size bins of the same species manifest divergent error trends (e.g., during the 8–10 day period, the slopes for different size bins are 0.00204 mg m$^{-2}$ h$^{-1}$, 0.00187 mg m$^{-2}$ h$^{-1}$, -0.000127 mg m$^{-2}$ h$^{-1}$, and -0.0000213 mg m$^{-2}$ h$^{-1}$, respectively). The emergence of this complex error variation is related to the dual influence governing each grid point errors in online continuous simulations: (1) The inherent limitations of the AIMACI scheme, which achieves accurate simulations when encountering well-learned input feature combinations, while exhibiting degraded performance under insufficiently trained input patterns; and (2) The compound error propagation mechanisms during continuous simulations, where input biases of species concentrations at each timestep are affected by three factors: local error accumulation from preceding steps, error

propagation through transport processes in numerical models from neighboring grids, and perturbations induced by other processes in numerical models (e.g., dry deposition and wet scavenging). Consequently, input biases exhibit nonlinear variability, with AIMACI's simulation accuracy being inversely correlated to input error magnitudes. In operational implementations of physics-AI hybrid models for online simulations, the influences of above two factors are often interrelated rather than independent, thereby amplifying the complexity of error variation. Introducing an error-correcting operators in the continuous simulation process may potentially enhance the stability of long-term simulations."

[Figure]

**Figure 3:** Sulfate column concentration simulations across different size bins. The first and second column depict the spatial distribution at the 10-day continuous simulation's end (2019-03-30 00:00 UTC), as simulated online by both the numerical

scheme and the AIMACI scheme. The third column is the absolute error between them. The fourth column shows the temporal evolution of the hourly RMSE over the 10-day period. The mean RMSE (unit: mg /m$^2$) for all days and the slope (unit: mg m$^{-2}$ h$^{-1}$) for different simulation stages (2-4day, 5-7day, 8-10day) are given inset.

● *11. Figure 4: These errors seem quite large, what are the effects of overestimating nitrate in this scheme? Are there more chemical reactions as a result? What are the chemical sinks of nitrate and MOSAIC and would this cause an error increase in other species?*

**Response:** Thank you for your insightful comment. You are correct that, if we only look at the relative error graph for nitrate (Figure 4h), we can see that the relative error at high altitudes is indeed large. We acknowledge that the original manuscript did not sufficiently explain this phenomenon. The observed behaviour arises because species concentrations at higher altitudes are typically below 1.0 μg/kg, and the relative error calculation divides the absolute error by these low concentration values. Consequently, even small absolute errors at high altitudes can result in significantly large relative errors. This effect is particularly pronounced for nitrate, which exhibits concentrations at higher altitudes that are one to two orders of magnitude lower than sulfate, often approaching zero. When considering the absolute error distribution (Figure 4g), the errors at higher altitudes are, in fact, relatively small.

However, we recognize that the AIMACI scheme does exhibit a tendency to overestimate nitrate concentrations. This overestimation can introduce biases in the input concentrations at each timestep during continuous simulations, potentially affecting the simulated concentrations of other species. When the overestimation is minor, its impact on other species is limited. In the atmosphere, the primary sinks of nitrate include: (1) Wet Deposition: Nitrate is removed from the atmosphere through precipitation (rain, snow); (2) Dry Deposition: Nitrate aerosols are deposited onto surfaces (e.g., vegetation, soil); (3) Photolysis and Re-release: Under specific conditions, nitrate aerosols may decompose and re-release NO$_x$. From the perspective of chemical sinks, underestimation of the photolysis process of nitrate could lead to overestimated nitrate concentrations. Additionally, from the perspective of chemical production, overestimation of the reaction between nitric acid and ammonia could also result in overestimated nitrate concentrations. However, in this study, our AIMACI scheme achieves end-to-end simulation of full aerosol chemistry and interactions. Therefore, it can't directly distinguish which process is causing the overestimation. Further investigation will be needed to identify the specific causes.

We have revised the relevant description of the manuscript to provide a clearer explanation of the significant relative errors observed for nitrate at high altitudes, as follows (Section 3.2.1, Lines 363-369):

"The absolute error exhibits a vertical gradient, with larger errors near the surface and smaller errors at higher altitudes. Conversely, the relative error distribution shows an inverse pattern, with lower relative errors near the surface and higher relative errors at elevated altitudes. This behavior arises because species concentrations at higher altitudes are typically below 1.0 μg/kg, and the relative error calculation divides the absolute error by these low concentration values. Consequently, even

small absolute perturbations at high altitudes can result in significantly large relative errors. This effect is particularly pronounced for nitrate, which exhibits concentrations at higher altitudes that are one to two orders of magnitude lower than sulfate, often approaching zero."

[Figure]

Figure 4: Zonal mean total concentrations (summed across 4 size bins) of sulfate and nitrate between 109.1 °E and 125.9 °E, as simulated online by both the numerical scheme and the AIMACI scheme. Results are averages over the entire 10-day simulation period.

● *12. The analysis with seasonal weather is nice, thank you for this type of discussion*

**Response:** Thank you for your positive feedback on the seasonal weather analysis.

● *13. Figure 3. Slope misspelled*

**Response:** Thank you for carefully reviewing the manuscript and identifying this typographical error. The misspelled label "Slpoe" in Figure 3 has been corrected to "Slope" in the revised manuscript.

[Figure]

**Figure 3:** Sulfate column concentration simulations across different size bins. The first and second column depict the spatial distribution at the 10-day continuous simulation's end (2019-03-30 00:00 UTC), as simulated online by both the numerical scheme and the AIMACI scheme. The third column is the absolute error between them. The fourth column shows the temporal evolution of the hourly RMSE over the 10-day period. The mean RMSE (unit: mg /m$^2$) for all days and the slope (unit: mg m$^{-2}$ h$^{-1}$) for different simulation stages (2-4day, 5-7day, 8-10day) are given inset.

- *14. "In the development of our AIMACI scheme, we faced a bifurcation of choices: whether to input all features of a single grid point and predict for that grid point individually, followed by iterating through all grid points, or to input all features for the entire 3D grid space simultaneously and predict for the entire 3D space at once. Most current AI large models such as Pangu (Bi et al., 2023) and Fengwu (Chen et al., 2023), opt for the latter approach, which inevitably requires the use of convolutional networks... "*

  *--> I am not sure why you do this, or your justification. Random forests and ANNs/LSTMs can work like this as you are basically creating a tabular data problem. Unless you are inserting some correlation between grids/points, I do not see the strength of using this MHSA. Maybe you can say that this approach is computationally simpler, but we know we can use FlashAttention or other techniques to alleviate such a bottleneck. It is true that ViTs perform very well, but without a counterfactual involved, I do not see the inherent benefit of using this ML model architecture unless you have tried others and failed.*

**Response:** Thank you for your comment. We acknowledge that the approach of creating training data based on individual grid points is algorithm-agnostic, meaning that other algorithms such as random forests, ANNs/LSTMs can also perform grid-based predictions.

Our intention in the original manuscript was to highlight the benefits of choosing to use individual grid points as training samples over using the entire 3D grid space as a single sample when developing an AI scheme coupled with a numerical model: (1) Larger Training Dataset: By using individual grid points, we can generate a much larger volume of training data since there are numerous grid points within a single 3D grid space at any given moment. (2) Flexibility in Simulation Domain Size: When the AI scheme is trained on individual grid points samples, it can be applied to simulate any size domains by iterating calculations for each grid point. In contrast, if the AI scheme is trained using the entire 3D grid space as a single sample, the shape of the input array to the AI scheme would be fixed to the size of the training sample. This would restrict the flexibility in setting the size of simulation domain such as the number and resolution of grids. (3) Reduced Computational Load: Opting for individual grid points generally requires less computation. If the entire 3D grid space is used as a single sample, convolutional networks are often necessary, which can be computationally intensive.

However, considering that current related studies predominantly use grid-based training data and that this discussion may appear somewhat abrupt in the context of the section, we have removed this statement from the revised manuscript to maintain coherence and focus.

- *15. "Despite these pronounced fluctuations"*

  *--> You did not address the concerns of the other reviewer regarding this point. You offer a good explanation but that should be put directly in the text. I do not need any direct comments from the authors addressed to me, everything should be in the text.*

**Response:** Thank you for your comment and sorry for not making the text clearer in the original manuscript. In the revised manuscript, we ensured that all responses were reflected in the text. In response to this comment, we have carefully reviewed our previous response and realized that it did not adequately address the concerns regarding the increase in fluctuation frequency of aerosol species concentrations after five days. The changes in local aerosol concentration are influenced by multiple factors, and the increased frequency of fluctuations may be related to the following reasons: (1) Variations in meteorological conditions, such as changes in wind speed and direction, can significantly influence the dispersion and transport of aerosols. Increased wind speed or frequent shifts in wind direction may lead to more pronounced concentration fluctuations. Additionally, changes in humidity can affect the wet scavenging and hygroscopic growth of aerosols. Precipitation events, in particular, can enhance the wet scavenging effect, leading to a significant reduction in aerosol concentrations. (2) Variations in anthropogenic emissions, such as differences in human activities during weekdays and weekends, can also contribute to changes in aerosol concentrations.

Given that the focus of this study is to evaluate the performance of the AIMACI scheme rather than to analyse the mechanisms behind the fluctuations in aerosol concentrations, we have not conducted a more detailed analysis to identify the specific causes. We have adjusted the relevant text to include a simple analysis of the generation of concentration fluctuations as follows (Lines 377-385):

"The occurrence of these pronounced fluctuations may be related to the following factors: (1) Variations in meteorological conditions, such as changes in wind speed and direction, can significantly influence the dispersion and transport of aerosols. Increased wind speed or frequent shifts in wind direction may lead to more pronounced concentration fluctuations. Additionally, changes in humidity can affect the wet scavenging and hygroscopic growth of aerosols. Precipitation events, in particular, can enhance the wet scavenging effect, leading to a significant reduction in aerosol concentrations. (2) Variations in anthropogenic emissions, such as differences in human activities during weekdays and weekends, can also contribute to changes in aerosol concentrations. A more detailed analysis would be required to identify the specific causes. Despite these pronounced fluctuations, the AIMACI scheme effectively reproduces these features without introducing systematic bias, achieving $R^2$ values larger than 0.97."

---

## Author Response (AR3)

**Editor**

*General comments:*

● *Public justification (visible to the public if the article is accepted and published):*
  *With the next revision, please remove supplement`s materials from your \*.pdf manuscript (pages 30-33), since the supplement is already presented itself in the system.*

Dear Editors:

We sincerely appreciate your editorial guidance and the opportunity to revise our manuscript. In response to your specific request regarding supplemental materials:

We have fully removed the supplementary content (original pages 30-33) from the main manuscript PDF file, as instructed. The supplemental materials remain exclusively in their designated file within the submission system, per the journal's requirements.

Thank you for your clear instructions and diligent review. Please let us know if any additional adjustments are needed to comply with the journal's formatting guidelines.

Yours sincerely,

Zihan Xia and co-authors